# Connectome-based reservoir computing with the `conn2res` toolbox

Laura E. Suárez[1,2], Agoston Mihalik [3], Filip Milisav [1], Kenji Marshall[4], Mingze Li[1,2], Petra E. Vértes [3], Guillaume Lajoie [2,5] & Bratislav Misic [1] ✉

The connection patterns of neural circuits form a complex network. How signaling in these circuits manifests as complex cognition and adaptive behaviour remains the central question in neuroscience. Concomitant advances in connectomics and artificial intelligence open fundamentally new opportunities to understand how connection patterns shape computational capacity in biological brain networks. Reservoir computing is a versatile paradigm that uses high-dimensional, nonlinear dynamical systems to perform computations and approximate cognitive functions. Here we present `conn2res`: an open-source Python toolbox for implementing biological neural networks as artificial neural networks. `conn2res` is modular, allowing arbitrary network architecture and dynamics to be imposed. The toolbox allows researchers to input connectomes reconstructed using multiple techniques, from tract tracing to noninvasive diffusion imaging, and to impose multiple dynamical systems, from spiking neurons to memristive dynamics. The versatility of the `conn2res` toolbox allows us to ask new questions at the confluence of neuroscience and artificial intelligence. By reconceptualizing function as computation, `conn2res` sets the stage for a more mechanistic understanding of structure-function relationships in brain networks.

Brains are complex networks of anatomically connected and functionally interacting neurons that have the ability to seamlessly assimilate and interact with a perpetually changing external environment[1]. Sensory stimuli elicit signaling events within structural connectivity networks and manifest as patterned neural activity. These emergent neural dynamics are thought to support the computations that underlie cognition and adaptive behavior. However, a computational framework that describes how information processing and functional specialization occur in brain networks remains elusive. Developing such a framework would require understanding the multiple levels of the information-processing hierarchy, from how the brain's network architecture shapes the complex activity patterns elicited by external stimuli, to how neural circuits extract from these evoked activity patterns the necessary information to compute with time-varying inputs.

How does network structure shape spatiotemporal patterns of neural activity, and how do neural dynamics support computations that underlie cognitive functions and behaviors? An important piece of the puzzle is the study of connectomics[2]. Technological and analytic advances in neuroimaging methods have made it possible to reconstruct the wiring patterns of nervous systems, yielding high-resolution connectomes of brains in multiple species[3–6]. The availability of connectomes has led to the formulation of a variety of models that aim to map network architecture to various functional aspects of the brain[7], such as emergent neural dynamics[8,9], functional co-activation patterns[10], and inter-individual differences in behavior[11–13]. Multiple network features are correlated with emergent functional phenomena[14–19], but there is no clear mechanistic link between the static network architecture and cognition.

[1]McConnell Brain Imaging Centre, Montréal Neurological Institute, McGill University, Montréal, QC, Canada. [2]Mila, Quebec Artificial Intelligence Institute, Montreal, QC, Canada. [3]Department of Psychiatry, University of Cambridge, Cambridge, UK. [4]Department of Bioengineering, Stanford University, Stanford, CA, USA. [5]Department of Mathematics and Statistics, Université de Montréal, Montreal, QC, Canada. ✉e-mail: bratislav.misic@mcgill.ca

Furthermore, descriptive studies of the connectome across different species provide evidence that structural connectivity networks display topological features that are thought to shape the segregation and integration of information[20]. For instance, the simultaneous presence of a highly clustered architecture of segregated modules promotes specialized information processing[21–27], while a densely interconnected core of high-degree hubs shortens communication pathways and promotes the integration of information from distributed specialized domains[28,29]. How these ubiquitous organizational principles of the architecture of the brain confer computational capacity remains unknown.

Artificial intelligence offers alternative ways to approach the link between structure and function in brain networks that take into account computation[30,31]. Within the expanding spectrum of artificial neural network models, reservoir computing makes it possible to describe how recurrent neural circuits extract information from a continuous stream of external stimuli and how they approximate complex time-varying functions[32,33]. In reservoir networks learning occurs exclusively at the readout connections, and hence the main network architecture of the reservoir does not require specific weight calibration, remaining fixed throughout training. This eliminates a confounder while avoiding biologically implausible credit assignment problems such as the use of backpropagation training[34]. These reasons make reservoir computing an ideal paradigm to study the effects of connectome architecture on computation and learning. In this regard, machine-learning and artificial intelligence algorithms offer new ways to study structure-function relationships in brain networks by conceptualizing function as a computational property[31,35].

Here we review the fundamentals of reservoir computing and how it can be applied to gain mechanistic insight about the information processing of biological neural circuits. We then present `conn2res` (https://github.com/netneurolab/conn2res), an open-source Python toolbox that implements connectomes as reservoirs to perform cognitive tasks. In the spirit of open-science, `conn2res` builds on top of and is interoperable with other third-party resources and research initiatives to offer an exhaustive set of experimental configurations/settings that researchers can experiment with. These include a comprehensive corpus of cognitive tasks spanning a wide spectrum of computational and behavioral paradigms, multiple local intrinsic dynamics, and various linear optimization algorithms for task learning. All of this combined with the possibility of implementing connectomes reconstructed at different scales and obtained from any imaging modality. We have added a tutorial section with several use-case examples to illustrate different types of inferences that the `conn2res` toolbox supports, as well as to showcase its flexibility in terms of network architecture, network dynamics, and task paradigm. While being inclusive of different modeling traditions, from microcircuits to whole-brain network models, `conn2res` contributes a novel way for researchers to explore the link between structure and function in biological brain networks.

## Building a reservoir computer

Reservoir computing (RC) is an umbrella term that unifies two computational paradigms, *liquid state machines*[32] and *echo-state networks*[36]. The two originated independently in the fields of computational neuroscience and machine-learning, respectively, with a common goal: exploiting the computational properties of complex, nonlinear dynamical systems[37]. However, the ideas encompassed by the RC paradigm had been around in different forms for more than two decades prior[38–40]. The conventional reservoir computing (RC) architecture consists of an input layer, followed by the reservoir and a readout module (Fig. 1a)[32,36,37]. Typically, the reservoir is a recurrent neural network (RNN) of nonlinear units, while the readout module is a simple linear model. The readout module is trained to read the activation states of the reservoir — elicited by an external input signal —

and map them to the desired target output in a supervised manner. In contrast to traditional artificial RNNs, the recurrent connections within the reservoir are fixed and randomly assigned; only the connections between the reservoir and the readout module are learned (Fig. 1a)[41].

So, how does RC work? RC capitalizes on the nonlinear response of high-dimensional dynamical systems, referred to as reservoirs. The reservoir performs a nonlinear projection of the input into a high-dimensional space. This transformation of the input converts nonlinearly separable signals into linearly separable ones such that a linear model in the readout module can be trained to map the transformed input to the desired output[41]. In other words, the reservoir converts inputs into rich dynamic patterns that contain integrated information about the history of inputs[4] and are read out linearly to solve complex tasks. As long as the reservoir has sufficient built-in dynamical complexity and rich dynamics, a large variety of input-output mappings can be realized, including the approximation of complex time-varying functions, such as forecasting chaotic time series, considered to be a problem of high computational complexity. Under certain conditions, such as the presence of fading memory and separation properties, reservoirs can act as universal function approximators[32,42–44].

The computational capabilities of the reservoir are thus determined by its dynamics, which arise from the interaction between the fixed network architecture of the reservoir and the equations or rules governing the time evolution of its internal units. Importantly, unlike traditional artificial neural networks (Fig. 2a), in RC the experimenter can specify the connectivity of the reservoir and the equations governing its local dynamics (Fig. 2b). Likewise, by tuning the parameters of the system, the experimenter can transition global network dynamics through qualitatively different dynamical regimes such as stability or chaos[45]. The RC paradigm thus offers the advantages that arbitrary network architectures and dynamics can be superimposed on the reservoir, providing a tool for neuroscientists to investigate how connectome organization and neural dynamics interact to support learning in biologically-informed reservoirs (Fig. 2c).

The architectural flexibility of RC is multi-scale: the network architecture of reservoirs can be informed by connectomes reconstructed at different spatial scales, from microcircuits to meso- and macro-scale networks (Fig. 1a). Depending on the context, the units of the reservoir represent either populations of neurons or entire brain regions. The choice of local dynamics is mainly determined by the spatial scale of the reservoir's network, but the nature of the research question at hand should also be considered (Fig. 1a). In contrast to traditional RNNs, in which global network dynamics are determined by connectivity changes due to learning, RC allows us to impose not only different types of local dynamics, but global dynamics governing the population-level behavior can also be controlled. This means that the dynamical regime — or qualitative dynamical behavior — of the reservoir can be tuned to progressively transition from stable to chaotic, thus passing through a critical phase transition, or *criticality*[46,47]. By parametrically tuning the dynamics to be closer or further from criticality, RC allows us to investigate the effects of qualitatively different neural trajectories near this critical point on the computational performance of the reservoir[47–49]. An additional advantage of the dynamical and structural flexibility of RC is the possibility to enforce computational priors in the form of either functional or structural inductive biases[50]. Therefore, RC allows us to explore the functional consequences of information-processing strategies, such as critical dynamics or the presence of computational priors, thought to be exploited by biological brains[45–47,51,52].

The fact that RC can be used with arbitrary network architectures and dynamics, plus the possibility of performing a variety of tasks spanning multiple cognitive domains — from perceptual-motor functions, memory and learning, to complex attention and executive functions — makes it ideal to investigate how specific network attributes and dynamics influence the neural computations that support

cognition[53]. Specifically, by implementing various tasks along these multiple cognitive domains, connectome-informed reservoirs allow us to systematically map network structure and dynamics to a unique set of identifiable computational properties exclusive to the task at hand (Fig. 1b). In this way, the RC framework allows us to build a comprehensive structural-functional ontology, relating network structure and dynamics to fundamental blocks of computation and, ultimately, to cognitive function.

The application of this hybrid approach between artificial intelligence and neuroscience goes beyond exploring the link between structure and function in the healthy brain. For instance, it can be applied in the clinical setting to study how neurological diseases affect learning in the network. By comparing the performance of reservoirs informed by clinical populations against those informed by healthy controls, this framework allows us to investigate whether cognitive decline, measured as variations in computational capacity, can be explained by measurable changes in network architecture due to neurodegeneration (Fig. 1c). Another relevant application of the RC

framework is the exploration of how the link between structure and function changes throughout adaptive processes such as development or evolution (Fig. 1c). For example, by implementing connectomes obtained throughout the lifespan or from different species, as reservoirs, this framework allows us to investigate how variations in network architecture translate into differences in computational capacity across ontogeny and phylogeny, respectively. In all cases, the RC framework allows for statistical significance testing by benchmarking empirical neural network architectures against random or null network models[54]. Altogether, this hybrid framework proposes a shift in the way structure-function relationships are studied in brain networks: from understanding function as a phenomenon (i.e., inter-regional functional interactions or functional activation maps), to a concept of function that is closer to the computational and information-processing properties of brain networks, thus contributing to a more mechanistic understanding of how computations and functional specialization emerge from the interaction between network architecture and dynamics in neural circuits.

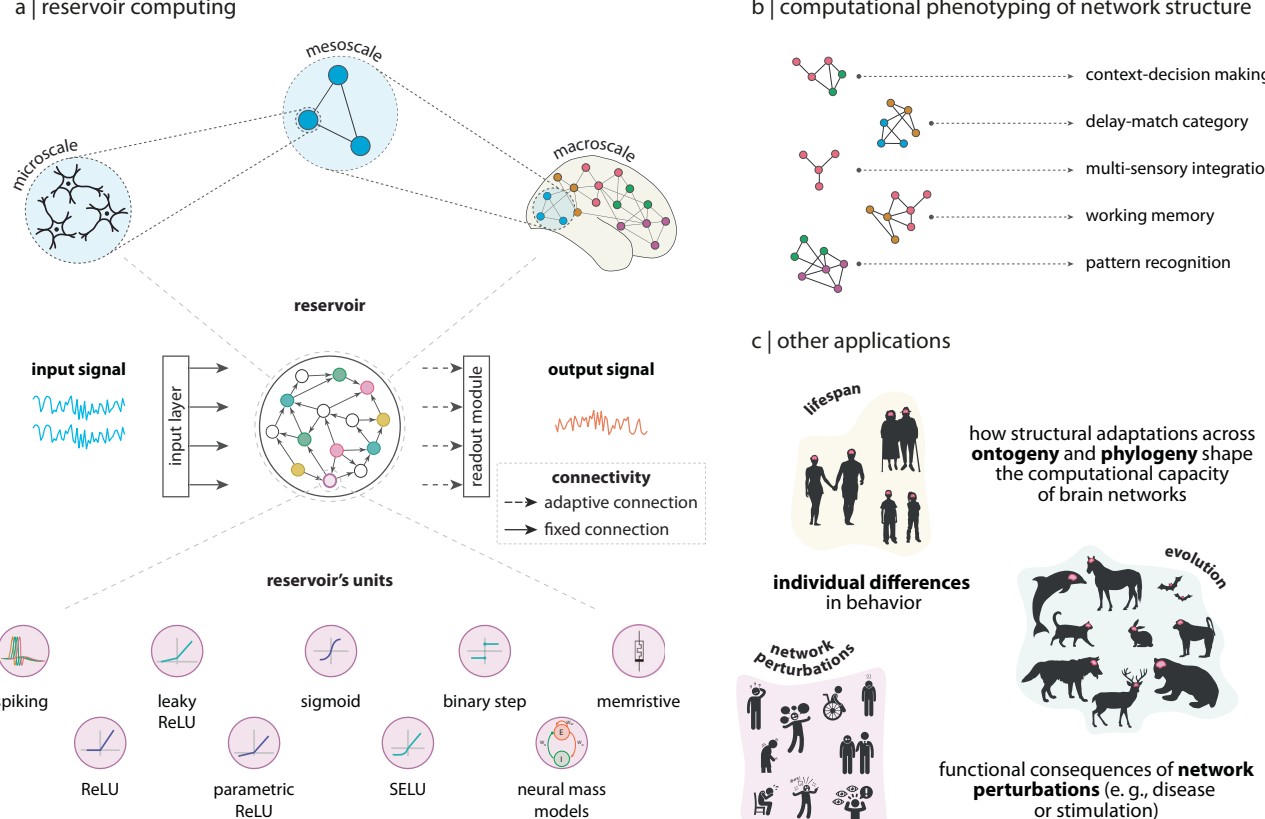

**Fig. 1 | Reservoir computing. a** The conventional reservoir computing architecture consists of an input layer, followed by a hidden layer, or *reservoir*, which is typically a recurrent neural network of nonlinear units, and the readout module, which is a simple linear model. In contrast to traditional artificial RNNs, the recurrent connections within the reservoir are fixed; only the connections between the reservoir and the readout module are trained. More importantly, RC allows arbitrary network architecture and dynamics to be implemented by the experimenter. Hence, biologically-plausible wiring patterns (top panel) and different types of local dynamics (bottom panel) can be superimposed on the reservoir. **b** By training connectome-informed reservoirs in a variety of tasks spanning multiple cognitive domains, we can systematically link network structure and dynamics to identifiable sets of computational properties. By doing so, we can build an extensive dictionary of structure-function relationships in which we relate brain network structure and dynamics to fundamental blocks of computation. **c** Other applications of this hybrid framework are for instance the investigation of how variations in connectome architecture support individual differences in computational capacity, or

the functional consequences of network perturbations due to pathology or external stimulation, or how structural adaptations across the lifespan or evolution shape the computational capacity of brain networks. In this way, the RC paradigm offers a tool for neuroscientists to investigate how network organization and neural dynamics interact to support learning in biologically-informed reservoirs. Credits: Young couple icon in panel (**c**) designed by Gordon Johnson from pixabay.com. Senior couple in panel (**c**) designed by Lizaveta Kadol from Vecteezy.com. Kids icon in panel (**c**) designed by clipart.me from FreeImages.com. Dolphin icon in panel (**c**) designed by Yulia Bulgakova from Vecteezy.com. Cat icon in panel (**c**) designed by gdakaska from pixabay.com. Wolf, rabbit, deer and bear icons in panel (**c**) designed by DesignsByOtto from Etsy.com. Horse, bat and macaque icons in panel (**c**) designed by svgsilh.com. "Disappointed", "crying", "crazy", "scolding", "disabled child", "sick" and "mental illness" icons in panel (**c**) designed by Gan Khoon Lay from thenounproject.com. Parkinson's disease icon in panel (**c**) designed by Peter van Driel from thenounproject.com. Paranoia icon in panel (**c**) designed by Adrien Coquet from thenounproject.com.

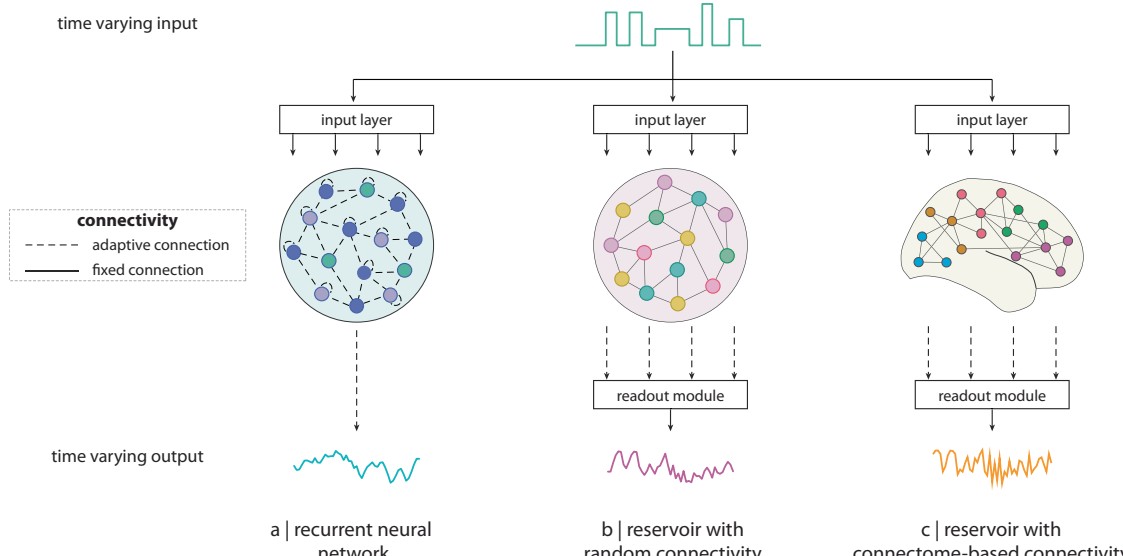

time varying input

input layer | input layer | input layer

connectivity
- - - - adaptive connection
──── fixed connection

readout module | readout module

time varying output

a | recurrent neural network

b | reservoir with random connectivity

c | reservoir with connectome-based connectivity

**Fig. 2 | The evolution of reservoir computing. a** Generic recurrent neural network (RNN) model. In classic RNNs, recurrent connections are learned via backpropagation-through-time[146]. The network topology that emerges from training does not necessarily result in biologically-plausible connectivity patterns. **b** The conventional reservoir computing architecture consists of a RNN with randomly assigned weights. The connections of the reservoir remain fixed during training and learning occurs only at the connections between the recurrent network and the readout module. Examples of this include classic liquid state machines[32] and echo-state-networks[36]. **c** Thanks to advances in imaging technologies, it is now possible to implement reservoirs with network architectures informed by empirical structural connectivity networks or connectomes. This allows us to explore the link between structure and function in biological brain networks from a computational point of view.

Apart from the advantages that RC offers to the neuroscience community, this paradigm is also promising from an engineering point of view. Reservoirs can be realized using physical systems, substrates or devices, as opposed to – generally time- and energy-consuming – simulated RNNs[55,56]. In this regard, the architecture of these neuromorphic chips could benefit from the emerging understanding of connection patterns in biological networks[57–59]. For instance, systematically mapping combinations of network attributes and dynamical regimes to a range of computational functions could assist the design of ad hoc or problem-specific tailored architectures. Due to their physical nature, neuromorphic systems are limited by spatial, material and energetic constraints, akin to biological neural networks. Because of this, insights gained about the economical organization of brain networks could contribute to the cost-effective design of these information-processing systems[35]. Furthermore, the fact that training only occurs at the readout stage makes RC an extraordinarily computationally efficient learning approach. In addition to this, parallel information processing can be achieved by simultaneously training multiple readout modules to perform various parallel tasks. Therefore, physical RC and RC in general, provide a powerful method for faster and simpler multi-task learning, compared to other RNNs. Thanks to the dynamical and versatile nature of the reservoir, the RC paradigm is perfectly suited for a wide range of supervised tasks involving the processing of temporal and sequential data. These include: time series prediction, dynamical pattern generation, classification and segmentation, control, signal processing, and monitoring of rare events, among others[60]. Because of all these reasons, physical RC systems have become ideal candidates for the development of novel brain-inspired computing architectures[61].

## RESULTS

### The conn2res toolbox

In a nutshell, conn2res is an open-source Python toolbox that allows users to implement biological neural networks as reservoirs to perform cognitive tasks (https://github.com/netneurolab/conn2res). The toolbox is built on top of the following well established, documented and maintained Python package dependencies: NumPy (https://numpy.org;[62–64]), SciPy (https://scipy.org;[65]), pandas (https://pandas.pydata.org;[66]), Scikit-Learn (https://scikit-learn.org;[67]), Gym (https://www.gymlibrary.dev;[68]), NeuroGym (https://neurogym.github.io;[53]), ReservoirPy (https://github.com/reservoirpy/reservoirpy;[69]), bctpy (https://github.com/aestrivex/bctpy;[70]), Seaborn (https://seaborn.pydata.org;[71]) and Matplotlib (https://matplotlib.org;[72]). The toolbox is also interoperable with other relevant Python packages including The Virtual Brain (https://www.thevirtualbrain.org;[73]), bio2art (https://github.com/AlGoulas/bio2art;[74]), NeuroSynth (https://neurosynth.org;[75]), Neuromaps (https://neuromaps-main.readthedocs.io;[76]) and the Enigma Toolbox (https://enigma-toolbox.readthedocs.io;[77]). Besides its extensive interoperability with other Python packages, a major strength of the conn2res toolbox is its flexibility in the choice of the different components that make part of the main RC workflow. The conn2res toolbox was expressly conceived as a tool for neuroscientists to explore a variety of hypotheses about structure-function coupling in brain networks. Therefore, compared to other RC-related packages, it offers higher flexibility in terms of network architecture, local dynamics, learning algorithms, task paradigms and performance metrics. To our knowledge, some of these are usually fixed or limited in other RC packages. Table S1 compares the conn2res toolbox against other well-known RC Python packages[69,78], based on these criteria.

The baseline conn2res workflow requires the following input arguments (Fig. 3a): (i) task name or dataset: the name of the task to be performed, or a labeled dataset of input-target pairs for supervised learning can also be provided. conn2res is a wrapper of NeuroGym[53], a curated collection of behavioral paradigms that were designed to facilitate the training of neural network models, and that are relevant for the neuroscience community. All of the 20+ tasks available in NeuroGym are also available in conn2res — some of these include perceptual decision making, context-dependent decision making, delayed comparison, delayed-paired association and delayed match category —; (ii) connectome: the connectivity matrix, which serves as the reservoir's network architecture. The toolbox supports binary and

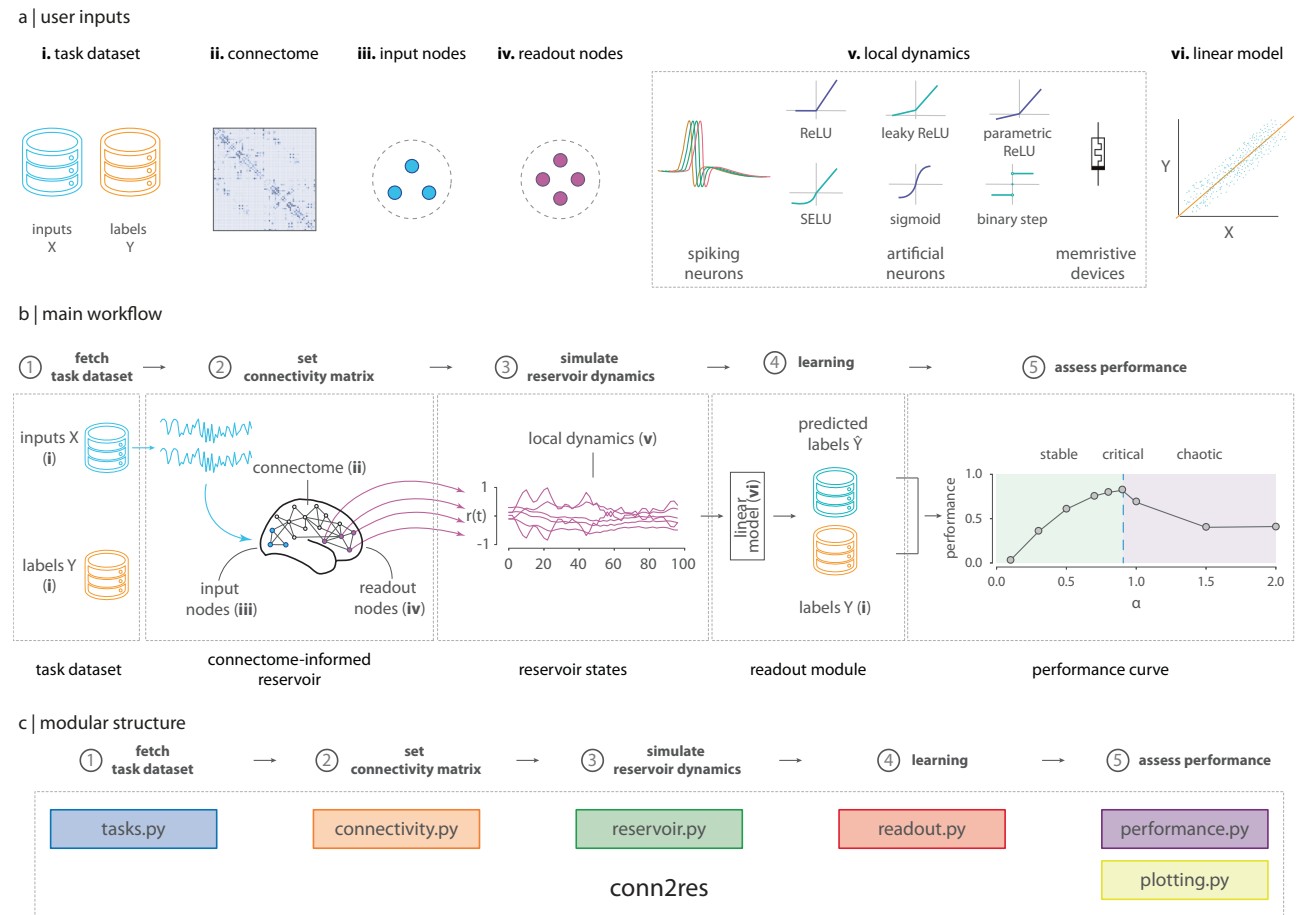

**Fig. 3 | conn2res toolbox. a** The general `conn2res` workflow requires the following parameters to be provided by the user: (i) a task name or a supervised learning dataset; (ii) a connectome or connectivity matrix; (iii) a set of input nodes; (iv) a set of readout nodes or modules; (v) the type of local dynamics, which can be either spiking neurons, artificial neurons (with a variety of activation functions), or memristive devices (for the simulation of physical reservoirs); and (vi) the linear model to be trained in the readout module. **b** In the mainstream `conn2res` workflow the input signal (X) is introduced to the reservoir through the input nodes (blue nodes). The signal propagates through the network, activating the states of the units within the reservoir. The activation states of the readout nodes (purple nodes) are then retrieved and used to train a linear model to approximate the target signal (Y). Depending on the type of the reservoir, the performance can be a single score or a curve of performance as a function of a parameter that tunes the dynamics of the reservoir[35]. **c** The `conn2res` toolbox has a modular architecture. It consists of six modules, each one comprising functions that support a specific step along the `conn2res` pipeline.

weighted connectivity matrices of both directed and undirected networks; (*iii*) input nodes: the set of nodes that receive the external input signals concerning the task at hand; (*iv*) readout nodes: the set of nodes or modules from which information will be retrieved to train the linear model in the readout module; (*v*) reservoir local dynamics: the type of dynamics governing the activation of the reservoir's units. Local dynamics can be split into two categories: discrete-time, governed by difference equations and continuous-time, based on differential equations. The former category includes both linear and nonlinear artificial neuron models with activation functions such as ReLU, leaky ReLU, sigmoid, and hyperbolic tangent, whereas the latter category includes a nonlinear spiking neuron model. The underlying leaky-integrate-and-fire neuron model is based on the framework proposed in[79], with model parameters from[80] and implements a double-exponential synaptic filter for the presynaptic spike trains. This flexible model provides higher biological plausibility, notably allowing for user-specified heterogeneous synaptic time constants, artificial stimulations/inhibitions, and interneuron-specific connectivity constraints[79,81]; (*vi*) linear model: specified as an instance of a linear model estimator from the `Scikit-Learn` library to be implemented for learning by the readout module[67].

The typical `conn2res` workflow is depicted in Fig. 3b. In the first stage, fetch task dataset, a supervised dataset consisting of input-label

pairs is either fetched from the `conn2res` repository, if the name of the task is provided by the user, or directly loaded if an external path is provided instead. In the second stage, set connectivity matrix, an instance of a reservoir object is created, and its network architecture and dynamics are set based on the connectivity matrix and the type of local nonlinearity specified by the user, respectively. In the third stage, simulate reservoir dynamics, the task inputs from the previous stage are introduced as external signals to the reservoir through the set of input nodes specified by the user. The dynamical models in the `conn2res` toolbox simulate the time evolution of the reservoir's internal units (which are activated thanks to the propagation of the external input signals), and generate time series activity for each node in the reservoir. In the fourth stage, learning, the time series activity of the readout nodes or modules specified by the user are retrieved and passed on to the readout module, together with the task labels from the first stage. Both of these are used to train the linear model in the readout module. Finally, during the fifth and last stage, assess performance, depending on the nature of the reservoir, the final output can be either a single performance score, or a performance curve that displays performance as a function of the parameter that controls for the qualitative behavior of the reservoir's dynamics (i.e., stable, critical or chaotic). Various performance metrics are currently available depending on whether the task requires a classification or a regression

model. To facilitate the user's experience, the toolbox provides several example scripts that illustrate use-case driven workflows.

The `conn2res` toolbox has a modular design. It consists of six modules, each one containing functions that support a specific step along the mainstream `conn2res` pipeline (Fig. 3c). The wrapper functions and classes used to generate the task datasets can be found in the *tasks.py* module. All types of manipulations on the connectivity matrix, such as binarization, weight scaling, normalization and rewiring, are handled by the Conn class in the *connectivity.py* module. Reservoir's features including its network architecture, local dynamics and the retrieval of the reservoir's activation states, are handled by the Reservoir class in the *reservoir.py* module. The functions in charge of the training and test of the linear model in the readout module are contained in the *readout.py* and *performance.py* modules, respectively. Finally, the *plotting.py* module offers a set of plotting tools that assist with the visualization of the different types of data generated along the pipeline, including the task input-output data, the 2D connectivity matrix of the reservoir's network architecture, the simulated reservoir states, the decision function of the readout module, and the performance curve.

## Tutorial

This section provides a broader overview of the multiple experimental settings and inferences that the `conn2res` toolbox supports. The first part consists of a detailed step-by-step example to illustrate the main `conn2res` workflow in action. The second part presents three applied cases in which specific hypotheses are proposed and tested using the toolbox. In each case, we evaluate the effect of global network architecture and dynamics on the computational capacity of reservoirs informed by connectomes of different animal species, reconstructed at different scales and obtained from different imaging modalities. Annotated notebooks and scripts to reproduce these results are included in the toolbox documentation https://github.com/netneurolab/conn2res/tree/master/examples.

**Example 1: toolbox components**. In this first example we quantify the effect of different types of local and global dynamics on the performance of a connectome-informed reservoir across two cognitive tasks: perceptual decision making[82] and context-dependent decision making[83]. To do so, we implement an echo-state network[36] whose connections are constrained by a human consensus connectome reconstructed from diffusion-weighted MRI data ($n = 66$ subjects. Data source: https://doi.org/10.5281/zenodo.2872624)[84]. To select the set of input and readout nodes, we use a functional connectivity-based partition of the connectome into intrinsic networks[85]. We define input nodes as a set of randomly selected brain regions from the visual system, and for the readout nodes we select all brain regions in the somatomotor system. Local dynamics are determined by the activation function of the reservoir's units. Here we use sigmoid and hyperbolic tangent activation functions. Global network dynamics are set by parametrically tuning $\alpha$, which corresponds to the spectral radius of the connectivity matrix[86]. The dynamics of the reservoir are considered to be stable if $\alpha < 1$, and chaotic if $\alpha > 1$. When $\alpha \approx 1$, the dynamics are said to be critical[46]. Because both tasks can be treated as supervised classification problems, we use a Ridge classifier model to train the readout module. We generate 1000 trials per task (70% training, 30% test), and we perform each task using 50 different realizations of the task labeled dataset. The distribution of performance scores is reported across the 50 instances of the task dataset.

Next we walk the reader through each of the steps along the main `conn2res` pipeline, and use the visualization tools included in the *plotting.py* module to depict the main output at each stage, facilitating the conceptual understanding of the workflow. Details about the practical implementation can be found in the *examples* folder of the `conn2res` toolbox. Results for the perceptual and context-dependent

decision-making tasks are shown on the left and right columns of Fig. 4, respectively. Top panel in Fig. 4 consists of a single plot that displays the time series of the input ($x_i$) and target labels ($y$) obtained during the task dataset fetching process. The perceptual decision-making task is a two-alternative forced choice task in which the reservoir must be able to integrate two stimuli to decide which one is higher on average (left column in Fig. 4). In the context-dependent decision-making task the reservoir has to perform one of two different perceptual discriminations, indicated by a contextual cue in every trial (right column in Fig. 4). Trials are delimited by vertical black dotted lines.

The second panel in Fig. 4 from top to bottom is a toy representation of the assignment of the connectome-based connectivity matrix to the reservoir's network (center); it also shows the input nodes (blue nodes on the left) used for the introduction of the external input signal into the reservoir during the simulation of the reservoir's dynamics, and the readout nodes (purple nodes on the right) used for the retrieval of information from the reservoir during the learning phase. The third panel in Fig. 4 depicts the simulation of the reservoir's dynamics and it consists of two plots: the top plot presents the time series of the input signals ($x$), while the bottom plot shows the simultaneous reservoir's activation state at every time step. These plots help the user visualize how reservoir states evolve as a function of the external inputs. The fourth panel in Fig. 4 makes reference to the learning process of the readout module during training. This panel contains three plots: the time series of the input signal (top), the decision function of the Ridge classifier (middle), and the predicted versus the ground truth target signals (bottom). Finally, the fifth panel in Fig. 4 shows the performance of the reservoir as a function of both local and global network dynamics. This panel presents two plots, each one corresponding to a different classification performance metrics: balanced accuracy (top) and F1 score (bottom). Each plot displays two curves that indicate how performance varies as a function of $\alpha$, and each curve corresponds to a different activation function: hyperbolic tangent (pink) and sigmoid (green).

Results in Fig. 4 suggest that both local and global network dynamics have an impact on task performance. At the local level, both tasks benefit from having a hyperbolic tangent activation function, compared to the sigmoid. However, dependence of task performance on global network dynamics varies from one task to the other. In the perceptual decision-making task, a choice must be made based on the time integration of two past stimuli, which means that a temporal memory is required. Because stability enforces memory in the reservoir, computations required in the perceptual decision-making task should take advantage of stable network dynamics[32,42,43,49]. This is indeed the case: if the local nonlinearity is hyperbolic tangent, a decrease in performance from stable ($\alpha < 1$) to chaotic ($\alpha > 1$) dynamics is observed (pink lines in bottom panel on the left column of Fig. 4). If the local nonlinearity is a sigmoid, however, the reservoir does not show a strong dependence with respect to global network dynamics (green lines in bottom panel on the left column of Fig. 4). In contrast, in the context-dependent decision-making task, a binary perceptual discrimination must be made, and hence the reservoir must learn to differentiate between two temporal patterns. Because chaotic dynamics contribute to the separability property of a reservoir[32,42,43,49], performance in this task should be enhanced by the presence of chaos. This is observed by an increase in performance as global network dynamics transition from stable to chaotic (pink and green lines in bottom panel on the right column of Fig. 4). Even though this is observed for both types of local nonlinearities - i.e., hyperbolic tangent and sigmoid - the effects are stronger for the hyperbolic tangent type. As expected, the effect of local and global network dynamics on task performance depends on the type of computations required by the task at hand.

This toy example helps us illustrate the flexibility of the `conn2res` toolbox in terms of choice of network architecture, local

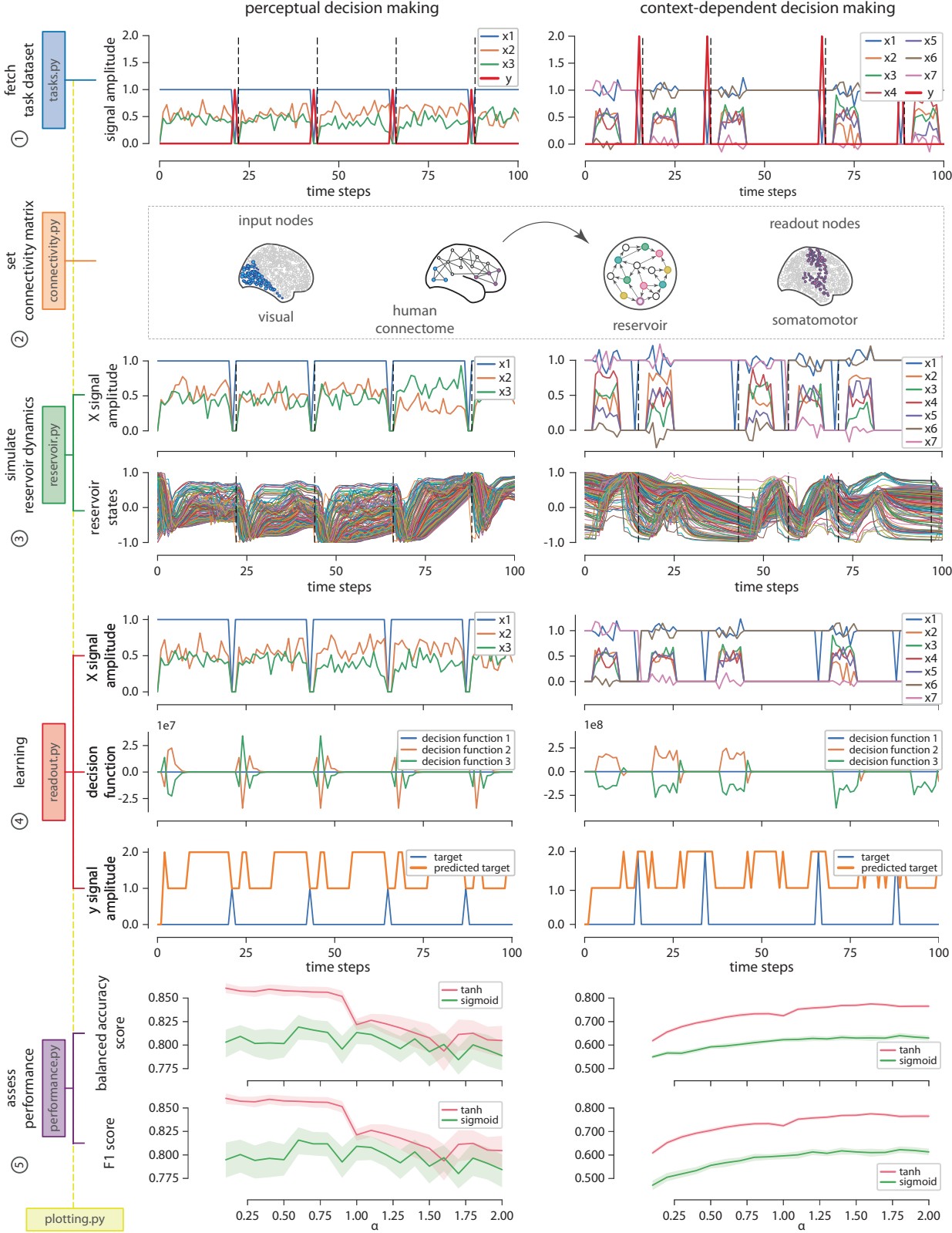

and global network dynamics, computational property, and performance metrics. Even though the type of experiments that the con-n2res toolbox has been designed for have more of an exploratory character, we expect that as imaging technologies improve together with our understanding of the anatomical structure of biological brains, more hypothesis-driven of experiments can be carried out with conn2res.

**Example 2: applications.** In the second part of the tutorial we show how the toolbox can be applied to address three specific biological questions. First, we quantify the memory capacity of human white matter connectomes reconstructed using diffusion-weighted MRI ($n = 66$;[84]). Here we use subcortical regions as input nodes and cortical regions as readout nodes. We then ask whether memory capacity specifically depends on the topological organization of the brain,

**Fig. 4 | Toolbox components.** Perceptual decision making (left column): is a two-alternative forced choice task ($y = \{1, 2\}$) in which the reservoir must be able to integrate two stimuli ($x_2$ and $x_3$; $x_1$ serves as a bias) to decide which one is higher on average. Context-dependent decision-making task (right column): in this task the reservoir has to perform one of two different perceptual discriminations ($y = \{1, 2\}$), indicated by a contextual cue in every trial (determined by $x_1$ to $x_7$). From top to bottom: the first panel displays the time series of the input ($x_i$) and target ($y$) signals obtained during the task dataset fetching step. The second panel presents a toy representation of the assignment of a connectome-based connectivity matrix to the reservoir's network (center). It also shows the set of input (left) and readout (right) nodes selected for the analysis[35]. The third panel displays the simulated reservoir's dynamics; the top plot shows the time series of the input signals and the bottom plot shows the simultaneous activation states of the readout nodes within the reservoir (results shown here correspond to the simulated reservoir states with hyperbolic tangent as activation function). The fourth panel illustrates the learning process that takes place in the readout module during training. At every time step, the top plot shows the input signals, the middle plot shows the decision function of the classifier --- in the readout module ---, and the bottom plot shows the predicted versus the target signal. Finally, the fifth panel shows the performance curves as a function of both local (hyperbolic tangent in pink and sigmoid in green) and global (varying the spectral radius $\alpha$ of the connectivity matrix) network dynamics. Two metrics were used to measure the performance of the classification: balanced accuracy (top) and F1 score (bottom). Solid lines represent mean performance across 500 iterations of the task dataset and shaded regions correspond to the 95% confidence interval.

rather than low-level features such as density or degree sequence. To address this question, we compare memory capacity in a group-level, empirical connectome against memory capacity in a population of 500 randomly rewired null connectomes with preserved density and degree sequence[54,87]. Figure 5a shows that at criticality ($\alpha = 1$), the memory capacity of empirical brain networks is significantly greater than in rewired nulls, suggesting that the topology of the human brain confers computational function[35].

In the previous example we focused on global computational capacity and showed that it relates to global network topology. For the second example, we demonstrate how the toolbox can be used to make inferences about regional heterogeneity or specificity for computational capacity. To address this question we implement the perceptual decision-making task on a single subject-level, connectome-informed reservoir. We stratify cortical nodes according to their affiliation with the canonical intrinsic networks[85]. Specifically, we use brain regions in the visual network as input nodes, and the remaining networks separately as a readout module each to quantify task performance. Figure 5b shows prominent differentiation in performance depending on which network is chosen as the readout module. Interestingly, the two modules with the greatest performance are the default mode and somatomotor networks, consistent with the notion that perceptual decision making involves integration of sensory inputs, comparison with internal goals, and formulation of an appropriate motor response[88]. Collectively, these results demonstrate how connectome-based reservoir computing can be used to make inferences about the computational capacity of anatomically circumscribed neural circuits.

For the final example, we show how the toolbox can be applied to comparative questions in which a researcher seeks to compare networks. In this example, we implement connectomes reconstructed from four different species: fruit fly[89], mouse[90], rat[91] and macaque[92]. As in the first example, we compare memory capacity in each empirical connectome with a population of 500 rewired null networks. Figure 5c shows that, despite differences in brain size, connectome resolution and reconstruction technique, the four model organism connectomes show a similar dependence on dynamics. Importantly, as with the human connectome, peak memory capacity is significantly greater in the empirical connectomes compared to the rewired nulls, except for the fruit fly, suggesting that this principle is potentially ubiquitous across nervous systems.

## Discussion

Despite common roots, modern neuroscience and artificial intelligence have followed diverging paths. The advent of high-resolution connectomics and the incredible progress of artificial neural networks in recent years present fundamentally new and exciting opportunities for the convergence of these vibrant and fast-paced fields. Here we briefly summarized the principles of the RC paradigm and introduced `conn2res`, an open-source code initiative designed to promote cross-pollination of ideas and bridge multiple disciplines,

including neuroscience, psychology, engineering, artificial intelligence, physics and dynamical systems. Below we look outward and propose how the `conn2res` toolbox can address emerging questions in these fields.

The `conn2res` toolbox embodies the versatility of the RC paradigm itself. By allowing arbitrary network architecture and dynamics to be superimposed on the reservoir, `conn2res` can be applied to investigate a wide range of neuroscience problems: from understanding the link between structure and function, studying individual differences in behavior, to exploring the functional consequences of network perturbations, such as disease or stimulation, or the computational benefits of specific architectural features, such as hierarchies and modules. The `conn2res` toolbox can readily accommodate network reconstructions at different spatial scales, from microcircuits to large-scale brain networks, and obtained using different imaging modalities, such as tract-tracing or diffusion MRI. Networks reconstructed at different points in either development and evolution can also be implemented in the toolbox to study, for instance, how structural adaptations across ontogeny and phylogeny shape computational capacity in brain networks. Collectively, `conn2res` offers new and numerous possibilities to discover how computation and functional specialization emerge from the brain's anatomical network structure.

The RC paradigm can also be adapted to jointly study the influence of network wiring and spatial embedding on computation. Namely, the placement of connections in the brain is subject to numerous material, energetic and spatial constraints, a factor that is often overlooked in classical paradigms that focus exclusively on network topology[93]. Right now the models included in the `conn2res` toolbox do not explicitly take into account spatial embedding but they can be readily adapted to do so. One way is to introduce conduction delays that are proportional to inter-regional connection lengths or geodesic distances over the cortical surface[94,95]. Another interesting and slightly different approach that incorporates geometric constraints is the recently introduced concept of spatially-embedded recurrent neural networks (seRNNs)[96]. These are recurrent networks with adaptive weights, confined within a 3D Euclidean space, whose learning is constrained by biological optimization processes, like the minimization of wiring costs or the optimization of inter-regional communicability, in addition to the maximization of computational performance. When the pruning of the network is guided by these biological optimization principles, the resulting network architecture displays characteristic features of biological brain networks, such as modular structure with a small-world topology, and the emergence of functionally specialized regions that are spatially co-localized and implement an energetically-efficient, mixed-selective code[96,97]. More broadly the cortex, which is typically studied in these models, is part of a wider network of the central nervous system that is embedded in a perpetually changing environment. The RC paradigm can accommodate this "embodied" view of the brain. Specifically, RC models can include adaptive homeostatic mechanisms that regulate

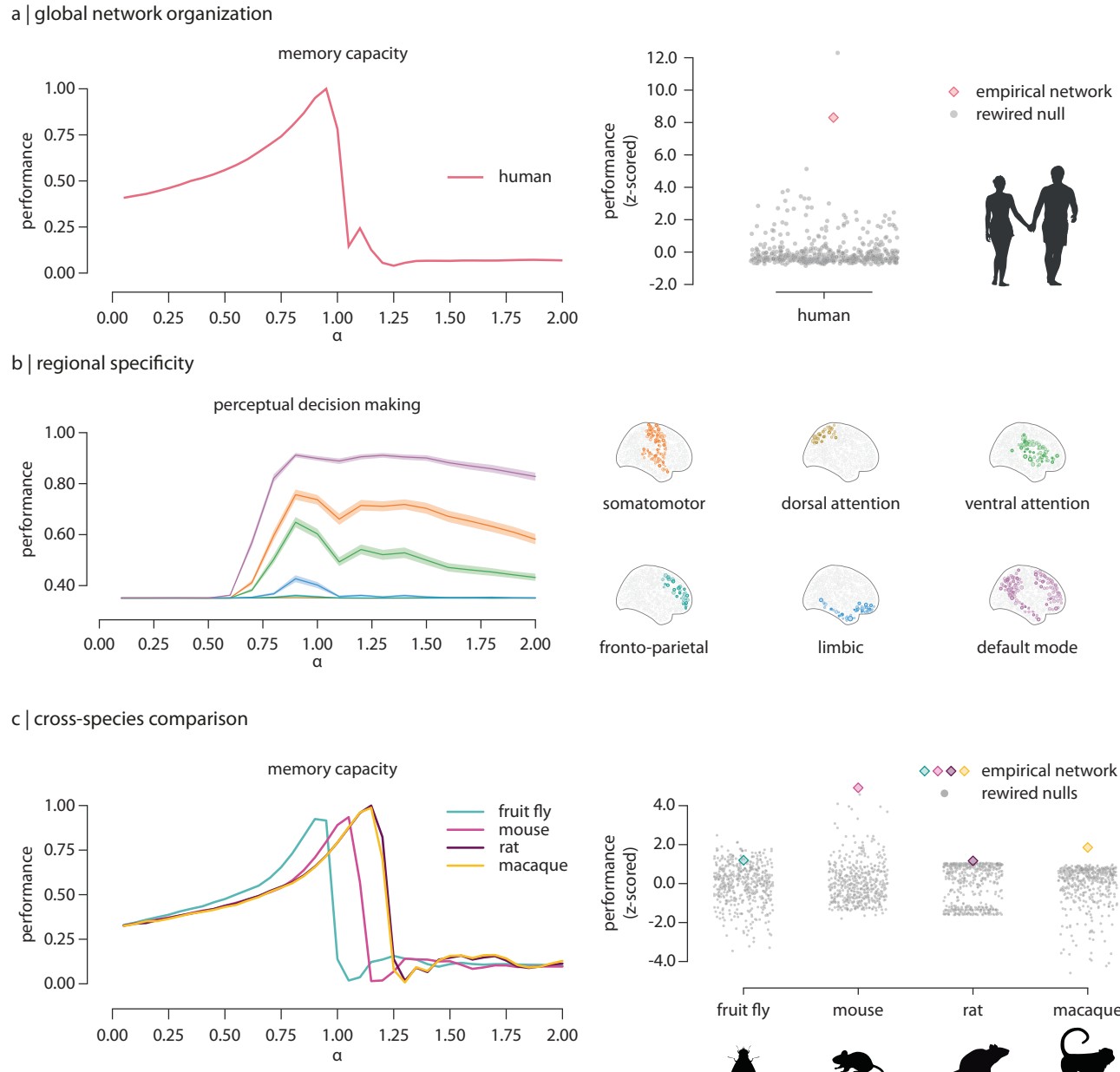

**Fig. 5 | Applied examples. a** Inferences on global network organization: a reservoir informed by a human consensus connectome ($n = 66$ subjects; diffusion-weighted MRI[84]) was implemented to perform a memory capacity task. Subcortical regions were used as input nodes while cortical regions were used as readout nodes. (Left) In all cases ---panels (**a**–**c**)---, global dynamics were tuned to transition from stable ($\alpha < 1$) to chaotic ($\alpha > 1$), where $\alpha$ corresponds to the spectral radius of the connectivity matrix. (Right) The performance of the empirical network was compared against the performance of a family of 500 rewired nulls that preserve network density and degree sequence[54,87]. At criticality ($\alpha_{crit} = 1$), empirical networks perform significantly better than rewired nulls ($p = 0.002$). **b** Anatomical inferences: a single, subject-level connectome-informed reservoir was implemented to perform a perceptual decision-making task (solid lines represent mean performance across 500 iterations of the task dataset; shaded regions correspond to the 95% confidence interval). (Right) Cortical regions were stratified according to the intrinsic network they belong to. Brain regions in the visual network were used as input nodes; the remaining intrinsic networks were used as separate readout modules each to quantify task performance. (Left) Across a wide range of $\alpha$ values ($\alpha > 0.5$), all intrinsic networks display significantly different behavior (one-way ANOVA $F = 1143.50$, $p < 0.002$ at $\alpha_{crit} = 1$), thus suggesting functional specialization across these networks. **c** Cross-species comparison: we implemented four distinct reservoirs, each informed by the connectome of a different model organism: fruit fly[89], mouse[90], rat[91] and macaque[92]. (Left) Connectome-informed reservoirs were trained to perform a memory capacity task. Sensory areas were used as input nodes while motor areas were used as readout nodes. As in (**a**), empirical networks were compared to a family of 500 rewired nulls[87]. At peak memory capacity, biologically-informed connectomes perform significantly better than rewired nulls, except for the fruit fly (fruit fly: $p = 0.11$ at $\alpha_{peak} = 0.9$, mouse: $p < 0.002$ at $\alpha_{peak} = 1.05$, rat: $p < 0.002$ at $\alpha_{peak} = 1.15$, macaque: $p < 0.002$ at $\alpha_{peak} = 1.15$). Credits: Young couple icon in panel (**a**) designed by Gordon Johnson from pixabay.com. Fruit fly, rat and monkey icons in panel (**c**) designed by Freepik.com. Mouse icon in panel (**c**) designed by CraftStarters.com.

brain-environmental feedback loops to ensure that reservoirs are maintained in a desired dynamical state such as criticality[98–100].

RC is often presented as a unified framework to train RNNs, but in a broader sense, it is a general framework to compute with high-dimensional, nonlinear dynamical systems, regardless of the choice of reservoir! Since any high-dimensional physical system with nonlinear dynamics could serve as reservoir − and these are abundant in both natural and man-made systems − a new field of applied research has emerged: physical reservoir computing. Here the goal is to exploit the rich dynamics of complex physical systems as information-processing devices. Physical substrates used for reservoirs are quite diverse: from analog circuits[101–104], field programmable gate arrays[105–108], photonic/opto-electronic devices[109–114], spintronics[115–117], quantum dynamics[118,119], nanomaterials[120–126], biological materials and organoids[127–133], mechanics and robotics[134–136], up to liquids or fluids[137,138], and most recently, origami structures[139]. The development of physical reservoir systems has been accompanied by advances in more efficient and effective RC frameworks, for instance by including time delays[140–142]. As physical reservoir computing becomes more popular, we envision the use of `conn2res` as a workbench to explore the effect of network interactions on the computational properties of physical reservoirs. Anticipating this, `conn2res` is currently equipped with a dedicated class for physical reservoirs, which allows memristive networks − a promising alternative for neuromorphic computing[122] − to be implemented as reservoirs. In this sense, the paradigm and the `conn2res` toolbox can be applicable to a wide variety of problems in adjacent scientific disciplines. From the neuro-connectomics perspective, `conn2res` offers new and numerous possibilities to discover how structure and function are linked in biological brain networks. From the artificial intelligence perspective, reverse-engineering biological networks will provide insights and novel design principles for re-engineering artificial, brain-inspired RC architectures and systems.

Altogether, `conn2res` is an easy-to-use toolbox that allows biological neural networks to be implemented as artificial neural networks. By combining connectomics and AI, the RC paradigm allows us to address new questions in a variety of scales of description and many adjacent fields. We hope that by reconceptualizing function as computation, `conn2res` allows us to take the next step towards understanding structure-function relationships in brain networks.

## Data availability
The structural human connectome data used in the first and second parts of the tutorial section of the present report are publicly available at https://doi.org/10.5281/zenodo.2872624[84]. The structural connectomes of the four model organisms used in the second part of the tutorial section are publicly available at: fruit fly (https://www.flycircuit.tw[89]), mouse (http://connectivity.brain-map.org[90]), rat (http://brancusi1.usc.edu/connections/grid/168[91]) and macaque (supporting information for https://www.pnas.org/doi/epdf/10.1073/pnas.1008054107[92,143]). To facilitate the reproduction of the results, all processed connectivity data used for the Tutorial section can be directly downloaded at https://doi.org/10.5281/zenodo.10205004[144].

## Code availability
Source code for `conn2res` is available on GitHub (https://github.com/netneurolab/conn2res) and is provided under the BSD 3-Clause "New" or "Revised" License. We have integrated `conn2res` with Zenodo (https://zenodo.org/doi/10.5281/zenodo.10437157[145]), which generates unique digital object identifiers (DOIs) for each new release of the toolbox. Researchers can access comprehensive online documentation via readthedocs (https://conn2res.readthedocs.io). Finally, as an open-source toolbox, `conn2res` is open to user suggestions and improvements, ensuring that it remains a continuously evolving resource. All code used for data processing, simulation, analysis, and figure generation relies on the following open-source Python packages: NumPy[62–64], Scipy[65], Pandas[66], Scikit-learn[67], bctpy https://github.com/aestrivex/bctpy[70], Gym https://github.com/openai/gym[68], NeuroGym https://github.com/neurogym/neurogym[53], Matplotlib[72], and Seaborn[71].

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

## Acknowledgements

We thank Bertha Vazquez-Rodriguez, Andrea Luppi, Ross Markello, Golia Shafiei, Vincent Bazinet, Justine Hansen, Zhen-Qi Liu, Eric Ceballos, Moohebat Pourmajidian and Asa Farahani for insightful comments on the manuscript. BM acknowledges support from the Natural Sciences and Engineering Research Council of Canada (NSERC Discovery Grant RGPIN #017-04265), from the Brain Canada Future Leaders Fund and the Canadian Institutes of Health Research (CIHR). GL acknowledges support from NSERC (Discovery Grant: RGPIN-2018-04821), CIFAR (Canada AI Chair), and a Canada Research Chair in Neural Computations and Interfacing (CIHR, tier 2). PEV acknowledges support from MQ: Transforming Mental Health (Grant No. MQF17-24). All research from the Department of Psychiatry at the University of Cambridge is made possible by the National Institute for Health and Care Research Cambridge Biomedical Research Centre and National Institute for Health and Care Research East of England Applied Research Centre. LES acknowledges support from the Fonds de Recherche du Québec - Nature et Technologies (FRQNT) Strategic Clusters Program (2020-RS4-265502 - Centre UNIQUE - Union Neurosciences and Artificial Intelligence - Quebec) and the Fonds de Recherche du Québec - Nature et Technologies (FRQNT).

## Author contributions

L.E.S., A.M., F.M., and B.M. conceived the toolbox. L.E.S. and B.M. wrote the manuscript, with valuable revision from P.E.V., A.M. and G.L. L.E.S. and A.M. developed the toolbox with help from K.M., M.L., and F.M. B.M. was the project administrator.

## Competing interests

The authors declare no competing interests.
