## [Peer Review File · Nature Communications]

REVIEWER COMMENTS

Reviewer #1 (Remarks to the Author):

This manuscript introduces an open-source Python toolbox, `conn2res`, designed to facilitate the transformation of biologically neuronal networks into artificial neural networks. With its modular architecture, `conn2res` offers the flexibility to implement various architectures and dynamics, enabling researchers to seamlessly integrate connectome data into reservoir network design, along with the choice of diverse dynamical systems.

The manuscript is structured into three distinct components:

1. A comprehensive review of reservoir computing fundamentals,
2. an exploration of the connections bridging reservoir computing and brain mechanisms, and
3. a comprehensive tutorial on utilizing the `conn2res` toolbox.

While the introduction of a toolbox that leverages connectome data for reservoir network design is intriguing, the manuscript's alignment with Nature Communications' publication criteria needs further clarification. Several concerns lead to the following assessment:

1. The three manuscript components, delineated as standalone, seem more directed towards framing the toolbox's context and potential, rather than delving deeply into its primary contributions.
2. The novelty of the work remains somewhat obscured by the extended literature review evident in the first two sections, coupled with the third section serving primarily as a tutorial for toolbox usage.
3. The toolbox's claimed functionalities appear unfulfilled as stated. For example, in the abstract, in the introductions, and in Fig. 1, the authors always claim that spiking neurons is one option for the internal dynamical system of the reservoir. This is inconsistent with the tutorial, which indicates that such features are yet to be implemented: "In the future, continuous time nonlinear dynamical models will be implemented, including spiking neuron models".

Given the aforementioned concerns, the manuscript might find a more appropriate fit within a specialized (bioinformatics-oriented) journal, where its contributions and limitations can be more comprehensively evaluated.

Reviewer #2 (Remarks to the Author):

In this manuscript, the authors present a modular Python toolbox that allows practitioners to implement biological-inspired artificial neural networks with arbitrary architectures. I agree with the authors in their claim about how the versatility of this tool can connect the areas of neuroscience and artificial intelligence towards efforts to understand the structure-function relationships in artificial neural networks and in the brain.

Although the manuscript is well written and does not present scientific errors, a considerable part of the text is structured like a big introduction or a review paper. This is good to introduce well the context of the work. However, I think the paper needs more details on methods and results in order to highlight its major contribution. For example, I miss details on how the connectome changes the structure of the reservoir and how these changes impact the results. These details are important to be shown and demonstrated with data in order to support the authors' biggest claim, which is the usage of this toolbox to understand the link between structure and function of artificial neural networks and the brain.

I really appreciate the hard work done by the authors in developing this toolbox with examples and tutorials and think this is very interesting for both neuroscience and artificial intelligence scientific communities. However, based on my comments above I recommend a revision of the manuscript before a resubmission to Nature Communications. Below, I also added some minor comments.

MINOR COMMENTS:

1. Despite the following text in the caption of Figure 1 "(d) ...reservoirs with connectome-based architectures to explore the link between structure and function in brain

networks from a computational point of view", It is not clear the difference between Fig.1(c) and Fig.1(d). Like I mention in my comment above, how the connectome-based connectivity changes the RC and how it changes the results?

2. In the second paragraph of the "Fundamentals of RC" section, more specifically in the sentence "... in RC the network architecture can be set and the dynamics can be tuned by the experimenter.", could the authors explain better what they mean by the dynamics can be tuned?

3. Missing reference in TUTORIAL section (last paragraph of page 8).

4. In Figure 4, the authors show how the global parameter affects the performance in two different tasks, but they do not explain why such behaviors happen or do not indicate a possible reason.

5. Legends and captions of figures are too small.

Reviewer #3 (Remarks to the Author):

The manuscript presents the conn2res toolbox for connectome constrained reservoir computing. It is very well written, including a very accessible presentation of the ideas behind reservoir computing and its history in computer science and neuroscience. I have explored the code which is implemented in a very clean but accessible way that should maximise its utility in the computational neuroscience community. I have some suggestions about additional topics/issues that should be addressed and about some simple additional tutorial examples that would extend the reach/accessibility of the toolbox.

The emphasis of the work is on connectome based models. Given the increasing recognition of geometric constraints on propagation of activity along the cortex, how would local geometric influences (e.g., dynamics driven by a mixture of geodesically proximal and white-matter connectome activity) be incorporated into models. Some discussion of this/computational tradeoffs would be useful.

There are other connectome-based reservoir-like models which haven't been mentioned in the manuscript (e.g., Hellyer et al, 2017 Plos Comp Biol). They also highlight other issues which could be touched on in the manuscript: e.g., brain-environmental feedback loops and adaptive, homeostatic mechanisms (see also Hellyer et al, 2016 Neuroimage). These may be relevant especially in the context of efficient/localised method for maintaining reservoirs near criticality; also potentially very relevant for modelling changes across time/disease, ontogeny/phylogeny. See also, Falandays et al 2023 Cognitive Neurodynamics for recent work (behavioural) on self-organisation to perform a range of tasks based on reservoir local homeostasis.

The tutorial workbook is very clean and accessible. Many users will just run the code with existing functions. However, it might be nice to have an example workbook of writing and running bespoke

models (e.g., other reservoirs/connectome approaches etc). This would make it much easier for users to adapt the code for a wider range of purposes.

Minor points:

"Given the sensitivity of Hebbian-like learning mechanisms to temporal relations [90–93], the synchronization of oscillatory activity can favour the formation of synaptic connections and hence can be used for establishing associations. In this way, as posited by the “binding by synchrony” hypothesis [94–96], synchronous oscillations act as a binding mechanism for distributed nodes in the network that represent features of the same cognitive object, thus serving as a general code of relatedness [96]. Likewise, phase shifts can lead to rapid reversals of the direction of information flow, thus serving as gating mechanisms for information transfer according to the “communication through coherence” hypothesis" I didn't understand the relevance of this paragraph.

"In the future, continuous-time nonlinear dynamical models will be implemented" Some expansion of this would be useful.

Dear Reviewers,

Thank you for the constructive feedback on our first submission, and for the opportunity to revise the manuscript. Following your comments and suggestions, we have thoroughly revised the manuscript. In this letter, we respond to each of the reviewers' comments in detail. Reviewer comments are in **bold font** and our responses are in regular font. If references to other works are used to support a response, these are included at the end of the response. New figures were appended with their respective response.

Major changes in the revised manuscript include:

- (1) *The brain as a reservoir* review section was entirely removed (~1,500 words). This streamlined the manuscript and optimized focus on the toolbox.
- (2) The *Fundamentals of Reservoir Computing* section was re-organized to make the narrative more user-friendly and directly tied to the toolbox.
- (3) The *Example 2: Applications* subsection was added to the *Tutorial* section with three detailed examples of how the toolbox can be used to address specific neurobiological questions. As with the other components, this is accompanied by ready-to-use example code.
- (4) We have added and tested spiking neural networks to the repertoire of dynamical systems.

We address each comment in detail below:

Reviewer #1 (Remarks to the Author):

This manuscript introduces an open-source Python toolbox, conn2res, designed to facilitate the transformation of biologically neuronal networks into artificial neural networks. With its modular architecture, conn2res offers the flexibility to implement various architectures and dynamics, enabling researchers to seamlessly integrate connectome data into reservoir network design, along with the choice of diverse dynamical systems.

The manuscript is structured into three distinct components:

- 1. A comprehensive review of reservoir computing fundamentals,**
- 2. an exploration of the connections bridging reservoir computing and brain mechanisms, and**
- 3. a comprehensive tutorial on utilizing the conn2res toolbox.**

While the introduction of a toolbox that leverages connectome data for reservoir network design is intriguing, the manuscript's alignment with Nature Communications' publication criteria needs further clarification. Several concerns lead to the following assessment:

We thank the Reviewer for the comments. Below we provide a detailed answer to each of the Reviewer's concerns.

- 1. The three manuscript components, delineated as standalone, seem more directed towards framing the toolbox's context and potential, rather than delving deeply into its primary contributions.**

This manuscript proposes both a framework and an open-source toolbox, both of which provide different contributions. The primary contribution of our hybrid framework, combining connectomics and reservoir computing, is the shift in the concept of brain function. Traditionally, structure-function relationships in the brain have been studied either through models that map structural connectivity to functional connectivity or other functional phenomena; or by mapping functional activation maps to specific cognitive functions. What we propose instead is a shift from a purely phenomenological concept of function to a concept that is closer to the computational and information-processing properties of biological brains. We have modified the *Fundamentals of Reservoir Computing* section of the manuscript to explicitly state this contribution:

“Altogether, this hybrid framework proposes a shift in the way structure-function relationships are studied in brain networks: from understanding function as a phenomenon (i.e., inter-regional functional interactions or functional activation maps), to a concept of function that is closer to the computational and information-processing properties of brain networks, thus contributing to a more mechanistic understanding of how computations and functional specialization

emerge from the interaction between network architecture and dynamics in neural circuits.”

Besides a conceptual shift, the proposed RC-based framework can be flexibly applied to address a variety of hypotheses about structure-function coupling in brain networks from a mechanistic standpoint. These applications range from building a comprehensive structural-functional ontology in the healthy brain, to exploring how network perturbations affect computation, to investigating how changes in network architecture across the lifespan or evolution shape the computational capacity of brain networks. These ideas were already contained in the manuscript, but we have modified Figure 1 to illustrate them:

Figure 1. Reservoir computing | (a) The conventional reservoir computing architecture consists of an input layer, followed by a hidden layer, or *reservoir*, which is typically a recurrent neural network of nonlinear units, and the readout module, which is a simple linear model. In contrast to traditional artificial RNNs, the recurrent connections within the reservoir are fixed; only the connections between the reservoir and the readout module are trained. More importantly, RC allows arbitrary network architecture and dynamics to be implemented by the experimenter. Hence, biologically-plausible wiring patterns (top panel) and different types of local dynamics (bottom panel) can be superimposed on the reservoir. (b) By training connectome-informed reservoirs in a variety of tasks spanning multiple cognitive domains, we can systematically link network structure and dynamics to identifiable sets of computational properties. By doing so, we can build an extensive dictionary of structure-function relationships in which we relate brain network structure and dynamics to fundamental blocks of computation. (c) Other applications of this hybrid framework are for instance the investigation of how variations in connectome architecture support individual differences in computational capacity, or the functional consequences of network perturbations due to pathology or external stimulation, or how structural adaptations across the lifespan or evolution shape the computational capacity of brain

networks. In this way, the RC paradigm offers a tool for neuroscientists to investigate how network organization and neural dynamics interact to support learning in biologically-informed reservoirs.

In addition, by building an open-source toolbox, we make available to the wide neuroscience community this NeuroAI framework. Despite the existence of other RC-related packages (e.g., *ReservoirPy* [1] or *echoes* [2]), the *conn2res* toolbox has been expressly conceived as a tool for neuroscientists. Namely, the options provided for parameters such as local dynamics or task paradigms in the *conn2res* toolbox are more suitable to address neuroscience-related questions. We have added a supplementary table in which we compare the *conn2res* toolbox to other RC Python packages in terms of the following criteria: network architecture, local dynamics, task paradigm, learning algorithm and performance metrics. We introduce this comparative table at the end of the first paragraph of *The conn2res toolbox* section:

“The *conn2res* toolbox was expressly conceived as a tool for neuroscientists to explore a variety of hypotheses about structure-function coupling in brain networks. Therefore, compared to other RC-related packages, it offers higher flexibility in terms of network architecture, local dynamics, learning algorithms, task paradigms and performance metrics. To our knowledge, some of these are usually fixed or limited in other RC packages. Table S1 compares the *conn2res* toolbox against other well-known RC Python packages [70, 79], based on these criteria.”

[70] Trouvain, N., Pedrelli, L., Dinh, T. T., & Hinaut, X. (2020, September). Reservoirpy: an efficient and user-friendly library to design echo state networks. In *International Conference on Artificial Neural Networks* (pp. 494-505). Cham: Springer International Publishing.

[79] Damicelli, F. *echoes: Echo State Networks with Python*; 2019.

	network architecture	local dynamics	task paradigms	learning algorithms	performance metrics
conn2res	Arbitrary	Artificial neuron models with different types of nonlinearities, spiking neurons (leaky-integrate-and-fire neurons) and memristive dynamics for neuromorphic devices (metastable switch model).	Includes classic RC tasks such as memory capacity, all behavioral paradigms included in the NeuroGym repository (e.g., context decision making, delay match category, delay	All linear models present in the Scikit-learn package are available.	Regression metrics: r-squared / coefficient of determination (R2), mean square error (MSE), root mean squared error (RMSE), normalized root mean squared error (NRMSE), mean absolute error (MAE), correlation coefficient. Classification metrics: accuracy, balanced accuracy, f1-score, precision and recall.

			match sample, interval discrimination, plus 15+ more), as well as all the non-neuroscience-related tasks included in the ReservoirPy package (e.g., prediction of chaotic time series such as Hénon map or Mackey-Glass time series) are also available.		
ReservoirPy	Arbitrary	Leaky integrator neurons	Tasks included are not necessarily neuroscience related (e.g., prediction of chaotic time series such as Hénon map or Mackey-Glass time series).	Logistic regression, Ridge classification and perceptron. Others include Recursive Least Squares and the Least Mean Squares learning rule.	Regression metrics: mean square error (MSE), root mean squared error (RMSE), normalized root mean squared error (NRMSE), r-squared / coefficient of determination (R2).
echoes	Arbitrary	Leaky integrator neurons	Prediction of chaotic Mackey-Glass time series.	Ridge regression	Regression metrics: r-squared / coefficient of determination (R2).
Table S1. The conn2res toolbox - comparison with other RC-based models					

Finally, to emphasize the applicability of the toolbox, we have modified the *Tutorial* section to include a total of four study-cases. The new *Tutorial* section is now divided in two parts: *Example 1: Toolbox components* and *Example 2: Applications*. The first part consists of a detailed step-by-step, toy example that illustrates the generic *conn2res* workflow with its graphic outputs, and also demonstrates its flexibility in terms of task paradigm, type of local nonlinearity and performance metric. The second part consists of three applied examples that show how the *conn2res* toolbox and more generally the framework can be used to answer questions of

biological interest. The new *Example 2: Applications* subsection includes a new figure (Figure 5). The new *Example 2: Applications* subsection in the *Tutorial* section reads:

“Example 2: Applications

In the second part of the tutorial we show how the toolbox can be applied to address three specific biological questions. First, we quantify the memory capacity of human white matter connectomes reconstructed using diffusion weighted MRI (n=66; [85]). Here we use subcortical regions as input nodes and cortical regions as readout nodes. We then ask whether memory capacity specifically depends on the topological organization of the brain, rather than low-level features such as density or degree-sequence. To address this question, we compare memory capacity in a group-level, empirical connectome against memory capacity in a population of 500 randomly rewired null connectomes with preserved density and degree sequence [55, 88]. Fig. 5a shows that at criticality ($\alpha = 1$), the memory capacity of empirical brain networks is significantly greater than in rewired nulls, suggesting that the topology of the human brain confers computational function [35].

In the previous example we focused on global computational capacity and showed that it relates to global network topology. For the second example, we demonstrate how the toolbox can be used to make inferences about regional heterogeneity or specificity for computational capacity. To address this question we implement the perceptual decision making task on a single subject-level, connectome-informed reservoir. We stratify cortical nodes according to their affiliation with the canonical intrinsic networks [86]. Specifically, we use brain regions in the visual network as input nodes, and the remaining networks separately as a readout module each to quantify task performance. Fig. 5b shows prominent differentiation in performance depending on which network is chosen as the readout module. Interestingly, the two modules with the greatest performance are the default mode and somatomotor networks, consistent with the notion that perceptual decision making involves integration of sensory inputs, comparison with internal goals, and formulation of an appropriate motor response [93]. Collectively, these results demonstrate how connectome-based reservoir computing can be used to make inferences about the computational capacity of anatomically circumscribed neural circuits.

For the final example, we show how the toolbox can be applied to comparative questions in which a researcher seeks to compare networks. In this example, we implement connectomes reconstructed from four different species: fruit fly [89], mouse [90], rat [91] and macaque [92]. As in the first example, we compare memory capacity in each empirical connectome with a population of 500 rewired null networks. Fig. 5c shows that, despite differences in brain size, connectome resolution and reconstruction technique, the four model organism connectomes show a similar dependence on dynamics. Importantly, as with the human

connectome, peak memory capacity is significantly greater in the empirical connectomes compared to the rewired nulls, except for the fruit fly, suggesting that this principle is potentially ubiquitous across nervous systems.”

a | global network organization

b | regional specificity

c | cross-species comparison

Figure 5. **Applied examples** | (a) Inferences on global network organization: a reservoir informed by a human consensus connectome ($n=66$ subjects; diffusion-weighted MRI [85]) was implemented to perform a memory capacity task. Subcortical regions were used as input nodes while cortical regions were used as readout nodes. (Left) In all cases ---panels (a) to (c)---, global dynamics were tuned to transition from stable ($\alpha < 1$) to chaotic ($\alpha > 1$). (Right) The performance of the empirical network was compared against the performance of a family of 500 rewired nulls that preserve network density and degree sequence [55, 88]. At criticality ($\alpha_{crit} = 1$), empirical networks perform significantly better than rewired nulls ($p = 0.002$). (b) Anatomical inferences: a single, subject-level connectome-informed reservoir was implemented to perform a perceptual decision making task (500 iterations). (Right) Cortical regions were stratified according to the intrinsic network they belong to. Brain regions in the

visual network were used as input nodes; the remaining intrinsic networks were used as separate readout modules each to quantify task performance. (Left) Across a wide range of α values ($\alpha > 0.5$), all intrinsic networks display significantly different behavior (one-way ANOVA $F = 1143.50$, $p < 0.002$ at $\alpha_{\text{crit}} = 1$), thus suggesting functional specialization across these networks. (c) Cross-species comparison: we implemented four distinct reservoirs, each informed by the connectome of a different model organism: fruit fly [89], mouse [90], rat [91] and macaque [92]. (Left) Connectome-informed reservoirs were trained to perform a memory capacity task. Sensory areas were used as input nodes while motor areas were used as readout nodes. As in (a), empirical networks were compared to a family of 500 rewired nulls [88]. At peak memory capacity, biologically-informed connectomes perform significantly better than rewired nulls, except for the fruit fly (fruit fly: $p = 0.11$ at $\alpha_{\text{peak}} = 0.9$, mouse: $p < 0.002$ at $\alpha_{\text{peak}} = 1.05$, rat: $p < 0.002$ at $\alpha_{\text{peak}} = 1.15$, macaque: $p < 0.002$ at $\alpha_{\text{peak}} = 1.15$).

[35] Suárez, L. E., Richards, B. A., Lajoie, G., & Misic, B. (2021). Learning function from structure in neuromorphic networks. *Nature Machine Intelligence*, 3(9), 771-786.

[55] Váša, F., & Mišić, B. (2022). Null models in network neuroscience. *Nature Reviews Neuroscience*, 23(8), 493-504.

[85] Griffa, A., Alemán-Gómez, Y., & Hagmann, P. (2019). Structural and functional connectome from 70 young healthy adults [data set]. *Zenodo*.

[86] Yeo, B. T., Krienen, F. M., Sepulcre, J., Sabuncu, M. R., Lashkari, D., Hollinshead, M., ... & Buckner, R. L. (2011). The organization of the human cerebral cortex estimated by intrinsic functional connectivity. *Journal of neurophysiology*.

[88] Maslov, S., & Sneppen, K. (2002). Specificity and stability in topology of protein networks. *Science*, 296(5569), 910-913.

[89] Chiang, A. S., Lin, C. Y., Chuang, C. C., Chang, H. M., Hsieh, C. H., Yeh, C. W., ... & Hwang, J. K. (2011). Three-dimensional reconstruction of brain-wide wiring networks in *Drosophila* at single-cell resolution. *Current biology*, 21(1), 1-11.

[90] Rubinov, M., Ypma, R. J., Watson, C., & Bullmore, E. T. (2015). Wiring cost and topological participation of the mouse brain connectome. *Proceedings of the National Academy of Sciences*, 112(32), 10032-10037.

[91] Bota, M., Sporns, O., & Swanson, L. W. (2015). Architecture of the cerebral cortical association connectome underlying cognition. *Proceedings of the National Academy of Sciences*, 112(16), E2093-E2101.

[92] Modha, D. S., & Singh, R. (2010). Network architecture of the long-distance pathways in the macaque brain. *Proceedings of the National Academy of Sciences*, 107(30), 13485-13490.

[93] Heekeren, H. R., Marrett, S., & Ungerleider, L. G. (2008). The neural systems that mediate human perceptual decision making. *Nature reviews neuroscience*, 9(6), 467-479.

References used in the response:

[1] Trouvain, N., Pedrelli, L., Dinh, T. T., & Hinaut, X. (2020, September). Reservoirpy: an efficient and user-friendly library to design echo state networks. In *International Conference on Artificial Neural Networks* (pp. 494-505). Cham: Springer International Publishing.

[2] Damicelli, F. *echoes: Echo State Networks with Python*; 2019.

2. The novelty of the work remains somewhat obscured by the extended literature review evident in the first two sections, coupled with the third section serving primarily as a tutorial for toolbox usage.

We agree with the Reviewer that the extensive literature review somewhat obscures the novelty of our work. We have taken several actions to shorten the manuscript and better highlight the novelty and contributions of our work. First, given its length, we decided to remove *The brain as a reservoir* review section from the manuscript and only include a few sentences at the beginning of the first paragraph of the *Fundamentals of Reservoir Computing* section to briefly describe the two main models that were originally listed under the umbrella term of reservoir computing:

“Reservoir computing (RC) is an umbrella term that unifies two computational paradigms, *liquid state machines* [32] and *echo-state networks* [36], originated independently in the fields of computational neuroscience and machine-learning, respectively, with a common goal: exploiting the computational properties of complex, nonlinear dynamical systems [37]. However, the ideas encompassed by the RC paradigm had been around in different forms for more than two decades prior [38-40]. [...]”

[32] Maass, W., Natschläger, T., & Markram, H. (2002). Real-time computing without stable states: A new framework for neural computation based on perturbations. *Neural computation*, 14(11), 2531-2560.

[36] Jaeger, H. (2001). The “echo state” approach to analysing and training recurrent neural networks-with an erratum note. *Bonn, Germany: German National Research Center for Information Technology GMD Technical Report*, 148(34), 13.

[37] Verstraeten, D., Schrauwen, B., d’Haene, M., & Stroobandt, D. (2007). An experimental unification of reservoir computing methods. *Neural networks*, 20(3), 391-403.

[38] Dominey, P. F., & Arbib, M. A. (1992). A cortico-subcortical model for generation of spatially accurate sequential saccades. *Cerebral cortex*, 2(2), 153-175.

[39] Dominey, P. F. (1995). Complex sensory-motor sequence learning based on recurrent state representation and reinforcement learning. *Biological cybernetics*, 73(3), 265-274.

[40] Dominey, P., Arbib, M., & Joseph, J. P. (1995). A model of corticostriatal plasticity for learning oculomotor associations and sequences. *Journal of cognitive neuroscience*, 7(3), 311-336.

Because the review section was removed, we modified Figure 2 accordingly. Panels c and d in the old version now correspond to panels b and c in the new version of the same Figure 2. A new reference was also added to the explanation of panel a. The new Figure 2 is:

Figure 2. **The evolution of reservoir computing** | (a) Generic recurrent neural network (RNN) model. In classic RNNs, recurrent connections are learned via backpropagation-through-time [46]. The network topology that emerges from training does not necessarily result in biologically plausible connectivity patterns. (b) The conventional reservoir computing architecture consists of a RNN with randomly assigned weights. The connections of the reservoir remain fixed during training and learning occurs only at the connections between the recurrent network and the readout module. Examples of this include classic liquid state machines [32] and echo-state-networks [36]. (c) Thanks to advances in imaging technologies, it is now possible to implement reservoirs with network architectures informed by empirical structural connectivity networks or connectomes. This allows us to explore the link between structure and function in biological brain networks from a computational point of view.

[46] Werbos, P. J. (1990). Backpropagation through time: what it does and how to do it. *Proceedings of the IEEE*, 78(10), 1550-1560.

[32] Maass, W., Natschläger, T., & Markram, H. (2002). Real-time computing without stable states: A new framework for neural computation based on perturbations. *Neural computation*, 14(11), 2531-2560.

[36] Jaeger, H. (2001). The “echo state” approach to analysing and training recurrent neural networks-with an erratum note. *Bonn, Germany: German National Research Center for Information Technology GMD Technical Report, 148(34)*, 13.

Other actions that were taken to highlight the novelty and contributions of our work are:

- The *Fundamentals of Reservoir Computing* section was modified to restate the conceptual paradigm shift that our combined framework proposes regarding how structure-function coupling is conceptualized and addressed in brain networks.
- Given the flexibility of the framework, it can be applied to address multiple aspects of the link between structure and function while providing mechanistic insight into how network topology and dynamics interact to shape computations in brain networks. These applications had already been mentioned in the manuscript, but we have modified Figure 1 to emphasize these opportunities.
- The *conn2res* toolbox offers certain specific advantages relevant for the neuroscience community that other RC-related Python packages do not. To emphasize this point, we had added a supplementary table in which we compare the *conn2res* toolbox with two other packages in terms of the alternatives offered for each of the components in the mainstream RC pipeline, such as local dynamics, task paradigm and performance metrics.
- Finally, we added three applied examples to the *Example 2: Applications* subsection of the *Tutorial* section that show how the *conn2res* toolbox can be used to answer questions of biological interest.

A detailed explanation for each of these points can be found in the response to Reviewer 1 ‘s first comment (pages 2 to 8 of the response letter).

3. The toolbox's claimed functionalities appear unfulfilled as stated. For example, in the abstract, in the introductions, and in Fig. 1, the authors always claim that spiking neurons is one option for the internal dynamical system of the reservoir. This is inconsistent with the tutorial, which indicates that such features are yet to be implemented: "In the future, continuous time nonlinear dynamical models will be implemented, including spiking neuron models".

We agree with the Reviewer that the manuscript was inconsistent and unclear regarding the availability of a spiking neuron model. We have therefore implemented such a model and revised the manuscript to indicate this clearly.

The new spiking neural network reservoir class is based on a leaky-integrate-and-fire neuron model with a double-exponential synaptic filter for the presynaptic spike trains with validated default parameters [1, 2], i.e., refractory time constant, membrane time constant, reset voltage, peak voltage, and rise time constant. The implementation was based on a framework for mapping trained, continuous-variable rate recurrent neural networks (RNNs) to spiking RNNs. We selected this model because it can be easily implemented as a reservoir and its integration

is interoperable with the existing *conn2res*'s modules and functionalities. Because the newly implemented model relies on reservoir computing principles, it does not require the training of a rate RNN as in the original framework. Instead, training is limited to the linear readout module as for the other *conn2res*'s models.

The spiking neuron model is very flexible. It allows for user-specified, neuron-wise synaptic time constants and time- and neuron-localized artificial stimulation or inhibition. Furthermore, for added biological detail, users can specify inhibitory neurons to enforce Dale's principle, i.e., ensuring that connections stemming from inhibitory neurons are strictly inhibitory and vice versa for excitatory connections, using a weight parameterization technique introduced in [3]. Moreover, inhibitory neurons can be further stratified into a somatostatin-expressing group (SST) and parvalbumin-positive group (PV) to constrain a common cortical microcircuit motif in which SST neurons inhibit both PV neurons and excitatory pyramidal neurons, but only receive excitatory afference [1].

The second paragraph of *The conn2res toolbox* section has been modified to reflect these changes:

"[...] **(v) reservoir local dynamics**: the type of dynamics governing the activation of the reservoir's units. Local dynamics can be split into two categories: discrete-time, governed by difference equations and continuous-time, based on differential equations. The former category includes both linear and non-linear artificial neuron models with activation functions such as ReLU, leaky ReLU, sigmoid, and hyperbolic tangent, whereas the latter category includes a non-linear spiking neuron model. The underlying leaky-integrate-and-fire neuron model is based on the framework proposed in [80], with model parameters from [81] and implements a double-exponential synaptic filter for the presynaptic spike trains. This flexible model provides higher biological plausibility, notably allowing for user-specified heterogeneous synaptic time constants, artificial stimulations/inhibitions, and interneuron-specific connectivity constraints [80, 82]; [...]"

[80] Kim, R., Li, Y., & Sejnowski, T. J. (2019). Simple framework for constructing functional spiking recurrent neural networks. *Proceedings of the national academy of sciences*, 116(45), 22811-22820.

[81] Nicola, W., & Clopath, C. (2017). Supervised learning in spiking neural networks with FORCE training. *Nature communications*, 8(1), 2208.

[82] Song, H. F., Yang, G. R., & Wang, X. J. (2016). Training excitatory-inhibitory recurrent neural networks for cognitive tasks: a simple and flexible framework. *PLoS computational biology*, 12(2), e1004792.

References used in the response:

[1] Kim, R., Li, Y., & Sejnowski, T. J. (2019). Simple framework for constructing functional spiking recurrent neural networks. *Proceedings of the national academy of sciences*, 116(45), 22811-22820.

[2] Nicola, W., & Clopath, C. (2017). Supervised learning in spiking neural networks with FORCE training. *Nature communications*, 8(1), 2208.

[3] Song, H. F., Yang, G. R., & Wang, X. J. (2016). Training excitatory-inhibitory recurrent neural networks for cognitive tasks: a simple and flexible framework. *PLoS computational biology*, 12(2), e1004792.

Given the aforementioned concerns, the manuscript might find a more appropriate fit within a specialized (bioinformatics-oriented) journal, where its contributions and limitations can be more comprehensively evaluated.

We thank the Reviewer for the previous suggestions. We have modified the manuscript in several ways such that the novelty and contributions of the proposed framework are more transparent to the reader. In particular, we agree with the Reviewer that the extended literature review section, namely *The brain as a reservoir*, somewhat obscures the contributions of our work and therefore we have decided to remove it altogether. We have also extended the *Tutorial* section with three additional examples that illustrate better the applicability of the toolbox.

Reviewer #2 (Remarks to the Author):

In this manuscript, the authors present a modular Python toolbox that allows practitioners to implement biological-inspired artificial neural networks with arbitrary architectures. I agree with the authors in their claim about how the versatility of this tool can connect the areas of neuroscience and artificial intelligence towards efforts to understand the structure-function relationships in artificial neural networks and in the brain.

Although the manuscript is well written and does not present scientific errors, a considerable part of the text is structured like a big introduction or a review paper. This is good to introduce well the context of the work. However, I think the paper needs more details on methods and results in order to highlight its major contribution. For example, I miss details on how the connectome changes the structure of the reservoir and how these changes impact the results. These details are important to be shown and demonstrated with data in order to support the authors' biggest claim, which is the usage of this toolbox to understand the link between structure and function of artificial neural networks and the brain.

I really appreciate the hard work done by the authors in developing this toolbox with examples and tutorials and think this is very interesting for both neuroscience and artificial intelligence scientific communities. However, based on my comments above I recommend a revision of the manuscript before a resubmission to Nature Communications. Below, I also added some minor comments.

We thank the Reviewer for the comments and suggestions. We agree with the Reviewer that a large part of the original manuscript was structured like a review paper and this shadows the contribution of our work. This is in line with Reviewer 1's major concern and therefore we have revised the manuscript in several ways to highlight its major contributions.

First, given its length and content we have decided to remove *The brain as a reservoir* review section. We just included a short paragraph at the beginning of the *Fundamentals of Reservoir Computing* section to briefly introduce the two main models that were originally listed under the umbrella term of reservoir computing:

“Reservoir computing (RC) is an umbrella term that unifies two computational paradigms, *liquid state machines* [32] and *echo-state networks* [36], originated independently in the fields of computational neuroscience and machine-learning, respectively, with a common goal: exploiting the computational properties of complex, nonlinear dynamical systems [37]. However, the ideas encompassed by the RC paradigm had been around in different forms for more than two decades prior [38-40]. [...]”

[32] Maass, W., Natschläger, T., & Markram, H. (2002). Real-time computing without stable states: A new framework for neural computation based on perturbations. *Neural computation*, 14(11), 2531-2560.

[36] Jaeger, H. (2001). The “echo state” approach to analysing and training recurrent neural networks-with an erratum note. *Bonn, Germany: German National Research Center for Information Technology GMD Technical Report*, 148(34), 13.

[37] Verstraeten, D., Schrauwen, B., d’Haene, M., & Stroobandt, D. (2007). An experimental unification of reservoir computing methods. *Neural networks*, 20(3), 391-403.

[38] Dominey, P. F., & Arbib, M. A. (1992). A cortico-subcortical model for generation of spatially accurate sequential saccades. *Cerebral cortex*, 2(2), 153-175.

[39] Dominey, P. F. (1995). Complex sensory-motor sequence learning based on recurrent state representation and reinforcement learning. *Biological cybernetics*, 73(3), 265-274.

[40] Dominey, P., Arbib, M., & Joseph, J. P. (1995). A model of corticostriatal plasticity for learning oculomotor associations and sequences. *Journal of cognitive neuroscience*, 7(3), 311-336.

Because the review section was removed, Figure 2 was modified accordingly. Panels c and d in the old version now correspond to panels b and c in the new version of the same Figure 2. A new reference was also added to the explanation of panel a. The new Figure 2 is:

Figure 2. **The evolution of reservoir computing** | (a) Generic recurrent neural network (RNN) model. In classic RNNs, recurrent connections are learned via backpropagation-through-time [46]. The network topology that emerges from training does not necessarily result in biologically plausible connectivity patterns. (b) The conventional reservoir computing architecture consists of a RNN with randomly assigned weights. The connections of the reservoir remain fixed during training and learning occurs only at the connections between the recurrent network and the readout module. Examples of this include classic liquid state machines [32] and echo-state-networks [36]. (c) Thanks to advances in imaging technologies, it is now possible to implement reservoirs with network architectures informed by empirical structural connectivity networks or connectomes. This allows us to explore the link between structure and function in biological brain networks from a computational point of view.

[46] Werbos, P. J. (1990). Backpropagation through time: what it does and how to do it. *Proceedings of the IEEE*, 78(10), 1550-1560.

[32] Maass, W., Natschläger, T., & Markram, H. (2002). Real-time computing without stable states: A new framework for neural computation based on perturbations. *Neural computation*, 14(11), 2531-2560.

[36] Jaeger, H. (2001). The “echo state” approach to analysing and training recurrent neural networks-with an erratum note. *Bonn, Germany: German National Research Center for Information Technology GMD Technical Report*, 148(34), 13.

In addition, following the Reviewer’s advice, we have extended the *Tutorial* section to include a total of four study-cases. The new *Tutorial* section is now divided in two parts: *Example 1: Toolbox components* and *Example 2: Applications*. The first part consists of a detailed step-by-step, toy example that illustrates the generic *conn2res* workflow with its graphic outputs, and also demonstrates its flexibility in terms of task paradigm, type of local nonlinearity and performance metric. The second part consists of three new applied examples that show how the *conn2res* toolbox can be used to address questions of biological relevance. Importantly, and as suggested by the Reviewer, the new examples evidence how different network topologies give rise to different computational performance, thus proving the convenience of *conn2res* as a tool to explore structure-function relationships in both biological and artificial neural networks. The scripts used for these examples have also been added to the toolbox (<https://github.com/netneurolab/conn2res/tree/master/examples>). The new *Example 2: Applications* subsection includes a new figure (Figure 5). The new *Example 2: Applications* subsection in the *Tutorial* section reads:

“Example 2: Applications

In the second part of the tutorial we show how the toolbox can be applied to address three specific biological questions. First, we quantify the memory capacity of human white matter connectomes reconstructed using diffusion weighted MRI (n=66; [85]). Here we use subcortical regions as input nodes and cortical regions as readout nodes. We then ask whether memory capacity specifically depends on the topological organization of the brain, rather than low-level features such as density or degree-sequence. To address this question, we compare memory capacity in a group-level, empirical connectome against

memory capacity in a population of 500 randomly rewired null connectomes with preserved density and degree sequence [55, 88]. Fig. 5a shows that at criticality ($\alpha = 1$), the memory capacity of empirical brain networks is significantly greater than in rewired nulls, suggesting that the topology of the human brain confers computational function [35].

In the previous example we focused on global computational capacity and showed that it relates to global network topology. For the second example, we demonstrate how the toolbox can be used to make inferences about regional heterogeneity or specificity for computational capacity. To address this question we implement the perceptual decision making task on a single subject-level, connectome-informed reservoir. We stratify cortical nodes according to their affiliation with the canonical intrinsic networks [86]. Specifically, we use brain regions in the visual network as input nodes, and the remaining networks separately as a readout module each to quantify task performance. Fig. 5b shows prominent differentiation in performance depending on which network is chosen as the readout module. Interestingly, the two modules with the greatest performance are the default mode and somatomotor networks, consistent with the notion that perceptual decision making involves integration of sensory inputs, comparison with internal goals, and formulation of an appropriate motor response [93]. Collectively, these results demonstrate how connectome-based reservoir computing can be used to make inferences about the computational capacity of anatomically circumscribed neural circuits.

For the final example, we show how the toolbox can be applied to comparative questions in which a researcher seeks to compare networks. In this example, we implement connectomes reconstructed from four different species: fruit fly [89], mouse [90], rat [91] and macaque [92]. As in the first example, we compare memory capacity in each empirical connectome with a population of 500 rewired null networks. Fig. 5c shows that, despite differences in brain size, connectome resolution and reconstruction technique, the four model organism connectomes show a similar dependence on dynamics. Importantly, as with the human connectome, peak memory capacity is significantly greater in the empirical connectomes compared to the rewired nulls, except for the fruit fly, suggesting that this principle is potentially ubiquitous across nervous systems.”

a | global network organization

b | regional specificity

c | cross-species comparison

Figure5. **Applied examples** | (a) Inferences on global network organization: a reservoir informed by a human consensus connectome ($n=66$ subjects; diffusion-weighted MRI [85]) was implemented to perform a memory capacity task. Subcortical regions were used as input nodes while cortical regions were used as readout nodes. (Left) In all cases ---panels (a) to (c)---, global dynamics were tuned to transition from stable ($\alpha < 1$) to chaotic ($\alpha > 1$). (Right) The performance of the empirical network was compared against the performance of a family of 500 rewired nulls that preserve network density and degree sequence [55, 88]. At criticality ($\alpha_{crit} = 1$), empirical networks perform significantly better than rewired nulls ($p = 0.002$). (b) Anatomical inferences: a single, subject-level connectome-informed reservoir was implemented to perform a perceptual decision making task (500 iterations). (Right) Cortical regions were stratified according to the intrinsic network they belong to. Brain regions in the visual network were used as input nodes; the remaining intrinsic networks were used as separate readout modules each to quantify task performance. (Left) Across a wide range of α values ($\alpha > 0.5$), all intrinsic networks display significantly different behavior (one-way ANOVA $F = 1143.50$, $p < 0.002$ at $\alpha_{crit} = 1$), thus suggesting functional specialization across these networks. (c) Cross-species

comparison: we implemented four distinct reservoirs, each informed by the connectome of a different model organism: fruit fly [89], mouse [90], rat [91] and macaque [92]. (Left) Connectome-informed reservoirs were trained to perform a memory capacity task. Sensory areas were used as input nodes while motor areas were used as readout nodes. As in (a), empirical networks were compared to a family of 500 rewired nulls [88]. At peak memory capacity, biologically-informed connectomes perform significantly better than rewired nulls, except for the fruit fly (fruit fly: $p = 0.11$ at $\alpha_{\text{peak}} = 0.9$, mouse: $p < 0.002$ at $\alpha_{\text{peak}} = 1.05$, rat: $p < 0.002$ at $\alpha_{\text{peak}} = 1.15$, macaque: $p < 0.002$ at $\alpha_{\text{peak}} = 1.15$).

[35] Suárez, L. E., Richards, B. A., Lajoie, G., & Misic, B. (2021). Learning function from structure in neuromorphic networks. *Nature Machine Intelligence*, 3(9), 771-786.

[55] Váša, F., & Mišić, B. (2022). Null models in network neuroscience. *Nature Reviews Neuroscience*, 23(8), 493-504.

[85] Griffa, A., Alemán-Gómez, Y., & Hagmann, P. (2019). Structural and functional connectome from 70 young healthy adults [data set]. *Zenodo*.

[86] Yeo, B. T., Krienen, F. M., Sepulcre, J., Sabuncu, M. R., Lashkari, D., Hollinshead, M., ... & Buckner, R. L. (2011). The organization of the human cerebral cortex estimated by intrinsic functional connectivity. *Journal of neurophysiology*.

[88] Maslov, S., & Sneppen, K. (2002). Specificity and stability in topology of protein networks. *Science*, 296(5569), 910-913.

[89] Chiang, A. S., Lin, C. Y., Chuang, C. C., Chang, H. M., Hsieh, C. H., Yeh, C. W., ... & Hwang, J. K. (2011). Three-dimensional reconstruction of brain-wide wiring networks in *Drosophila* at single-cell resolution. *Current biology*, 21(1), 1-11.

[90] Rubinov, M., Ypma, R. J., Watson, C., & Bullmore, E. T. (2015). Wiring cost and topological participation of the mouse brain connectome. *Proceedings of the National Academy of Sciences*, 112(32), 10032-10037.

[91] Bota, M., Sporns, O., & Swanson, L. W. (2015). Architecture of the cerebral cortical association connectome underlying cognition. *Proceedings of the National Academy of Sciences*, 112(16), E2093-E2101.

[92] Modha, D. S., & Singh, R. (2010). Network architecture of the long-distance pathways in the macaque brain. *Proceedings of the National Academy of Sciences*, 107(30), 13485-13490.

[93] Heekeren, H. R., Marrett, S., & Ungerleider, L. G. (2008). The neural systems that mediate human perceptual decision making. *Nature reviews neuroscience*, 9(6), 467-479.

Further actions were taken to highlight the novelty and contributions of our work:

The *Fundamentals of Reservoir Computing* section was modified to explicitly declare the conceptual paradigm shift that our combined framework proposes regarding how structure-function coupling is conceptualized and addressed in brain networks:

“Altogether, this hybrid framework proposes a shift in the way structure-function relationships are studied in brain networks: from understanding function as a phenomenon (i.e., inter-regional functional interactions or functional activation maps), to a concept of function that is closer to the computational and information-processing properties of brain networks, thus contributing to a more mechanistic understanding of how computations and functional specialization emerge from the interaction between network architecture and dynamics in neural circuits.”

We modified Figure 1 to illustrate how our combined framework can be applied to address multiple structure-function hypotheses in brain networks, while providing mechanistic insights into how network topology and dynamics interact to shape computations in brain networks:

Figure 1. Reservoir computing | (a) The conventional reservoir computing architecture consists of an input layer, followed by a hidden layer, or *reservoir*, which is typically a recurrent neural network of nonlinear units, and the readout module, which is a simple linear model. In contrast to traditional artificial RNNs, the recurrent connections within the reservoir are fixed; only the connections between the reservoir and the readout module are trained. More importantly, RC allows arbitrary network architecture and dynamics to be implemented by the experimenter. Hence, biologically-plausible wiring patterns (top panel) and different types of local dynamics (bottom panel) can be superimposed on the reservoir. (b) By training connectome-informed reservoirs in a variety of tasks spanning multiple cognitive domains, we can systematically link network structure and dynamics to identifiable sets of computational properties. By doing so, we can build an extensive dictionary of structure-function relationships in which we relate brain network structure and dynamics to fundamental blocks of

computation. (c) Other applications of this hybrid framework are for instance the investigation of how variations in connectome architecture support individual differences in computational capacity, or the functional consequences of network perturbations due to pathology or external stimulation, or how structural adaptations across the lifespan or evolution shape the computational capacity of brain networks. In this way, the RC paradigm offers a tool for neuroscientists to investigate how network organization and neural dynamics interact to support learning in biologically-informed reservoirs.

Because the *conn2res* toolbox offers certain specific advantages relevant for the neuroscience community that other RC-related Python packages do not, we added a supplementary table to emphasize this point. The table shows a comparison between *conn2res* and two other packages in terms of the alternatives offered for each of the components in the mainstream RC pipeline, such as local dynamics, task paradigm and performance metrics. The comparative table was introduced at the end of the first paragraph of *The conn2res toolbox* section:

“The *conn2res* toolbox was expressly conceived as a tool for neuroscientists to explore a variety of hypotheses about structure-function coupling in brain networks. Therefore, compared to other RC-related packages, it offers higher flexibility in terms of network architecture, local dynamics, learning algorithms, task paradigms and performance metrics. To our knowledge, some of these are usually fixed or limited in other RC packages. Table S1 compares the *conn2res* toolbox against other well-known RC Python packages [70, 79], based on these criteria.”

[70] Trouvain, N., Pedrelli, L., Dinh, T. T., & Hinaut, X. (2020, September). Reservoirpy: an efficient and user-friendly library to design echo state networks. In *International Conference on Artificial Neural Networks* (pp. 494-505). Cham: Springer International Publishing.

[79] Damicelli, F. *echoes: Echo State Networks with Python*; 2019.

	network architecture	local dynamics	task paradigms	learning algorithms	performance metrics
conn2res	Arbitrary	Artificial neuron models with different types of nonlinearities, spiking neurons (leaky-integrate-and-fire neurons) and memristive dynamics for neuromorphic devices (metastable switch model).	Includes classic RC tasks such as memory capacity, all behavioral paradigms included in the NeuroGym repository (e.g., context making, delay match category, delay	All linear models present in the Scikit-learn package are available.	Regression metrics: r-squared / coefficient of determination (R ²), mean square error (MSE), root mean squared error (RMSE), normalized root mean squared error (NRMSE), mean absolute error (MAE), correlation coefficient. Classification metrics: accuracy, balanced accuracy, f1-score, precision and recall.

			match sample, interval discrimination, plus 15+ more), as well as all the non-neuroscience-related tasks included in the ReservoirPy package (e.g., prediction of chaotic time series such as Hénon map or Mackey-Glass time series) are also available.		
ReservoirPy	Arbitrary	Leaky integrator neurons	Tasks included are not necessarily neuroscience related (e.g., prediction of chaotic time series such as Hénon map or Mackey-Glass time series).	Logistic regression, Ridge classification and perceptron. Others include Recursive Least Squares and the Least Mean Squares learning rule.	Regression metrics: mean square error (MSE), root mean squared error (RMSE), normalized root mean squared error (NRMSE), r-squared / coefficient of determination (R2).
echoes	Arbitrary	Leaky integrator neurons	Prediction of chaotic Mackey-Glass time series.	Ridge regression	Regression metrics: r-squared / coefficient of determination (R2).
Table S1. The conn2res toolbox - comparison with other RC-based models					

MINOR COMMENTS:

1. Despite the following text in the caption of Figure 2 "(d) ...reservoirs with connectome-based architectures to explore the link between structure and function in brain networks from a computational point of view", It is not clear the difference between Fig.2(c) and Fig.2(d). Like I mention in my comment above, how the connectome-based connectivity changes the RC and how it changes the results?

Figure 2 was modified to accommodate a more streamlined version of the manuscript (i.e., removing *The brain as a reservoir* review section). Panels c and d in the old version now correspond to panels b and c in the new version of the same Figure 2.

To answer the Reviewer’s question, the main difference between panels b (reservoir with random connectivity) and c (reservoir with connectome-based connectivity) lies in the connectivity patterns used for the reservoir. Classic reservoir computing methods make use of random connectivity matrices. What *conn2res* proposes instead is to make use of empirical connectomics data, which consists of biologically-plausible connectivity patterns, to define the network architecture of the reservoir. We have revised the caption of Figure 2 to clarify these differences between panels c and d, which now correspond to panels b and c. The new Figure and caption are as follows:

Figure 2. The evolution of reservoir computing | (a) Generic recurrent neural network (RNN) model. In classic RNNs, recurrent connections are learned via backpropagation-through-time [46]. The network topology that emerges from training does not necessarily result in biologically plausible connectivity patterns. (b) The conventional reservoir computing architecture consists of a RNN with randomly assigned weights. The connections of the reservoir remain fixed during training and learning occurs only at the connections between the recurrent network and the readout module. Examples of this include classic liquid state machines [32] and echo-state-networks [36]. (c) Thanks to advances in imaging technologies, it is now possible to implement reservoirs with network architectures informed by empirical structural connectivity networks or connectomes. This allows us to explore the link between structure and function in biological brain networks from a computational point of view.

[46] Werbos, P. J. (1990). Backpropagation through time: what it does and how to do it. *Proceedings of the IEEE*, 78(10), 1550-1560.

[32] Maass, W., Natschläger, T., & Markram, H. (2002). Real-time computing without stable states: A new framework for neural computation based on perturbations. *Neural computation*, 14(11), 2531-2560.

[36] Jaeger, H. (2001). The “echo state” approach to analysing and training recurrent neural networks-with an erratum note. *Bonn, Germany: German National Research Center for Information Technology GMD Technical Report*, 148(34), 13.

Furthermore, the first and third study cases of the new *Example 2: Applications* subsection of the *Tutorial* section are a good example of how biologically-informed connectomes can yield better performance in a memory capacity task compared to a distribution of randomly rewired networks (Fig. 5a and Fig. 5c).

2. In the second paragraph of the "Fundamentals of RC" section, more specifically in the sentence "... in RC the network architecture can be set and the dynamics can be tuned by the experimenter.", could the authors explain better what they mean by the dynamics can be tuned?

We agree with the Reviewer that it is not very clear what the expression “the dynamics can be tuned but the experimenter” means. What we mean by this is that the experimenter can arbitrarily select the dynamical model or equations that govern the time evolution of the reservoir’s local units, and also tune the parameters of the system to transition between qualitatively distinct global dynamical regimes such as stability or chaos. We have revised the manuscript to clarify this:

“[...] Importantly, unlike traditional artificial neural networks (Fig. 2a), in RC the experimenter can specify the connectivity of the reservoir and the equations governing its local dynamics (Fig. 2b). Likewise, by tuning the parameters of the system, the experimenter can transition global network dynamics through qualitatively different dynamical regimes such as stability or chaos [45]. [...]”

[45] Deco, G., & Jirsa, V. K. (2012). Ongoing cortical activity at rest: criticality, multistability, and ghost attractors. *Journal of Neuroscience*, 32(10), 3366-3375.

3. Missing reference in TUTORIAL section (last paragraph of page 8).

Thank you for catching this. We have added a reference to the MRI dataset we got the human data from. The reference is:

[85] Griffa, A., Alemán-Gómez, Y., & Hagmann, P. (2019). Structural and functional connectome from 70 young healthy adults [data set]. *Zenodo*.

4. In Figure 4, the authors show how the global parameter affects the performance in two different tasks, but they do not explain why such behaviors happen or do not indicate a possible reason.

We agree with the Reviewer that an explanation about why global parameters affect performance in the way they do in the proposed tasks is missing. A possible reason to explain this behavior are the different types of computations required by both tasks and the fact that these computations are supported by distinct global network dynamics, e.g., whether the activity of the reservoir is stable or chaotic [1].

In the perceptual decision making task the reservoir should make a decision based on the temporal integration of previous stimuli, which means that the reservoir needs to have some sort of temporal memory. Previous studies on the computational capacity of reservoirs have shown that stable dynamics support the memory capacity of the network. Therefore, a decrease in performance would be expected when the dynamics of the reservoir transition from stable ($\alpha < 1$) to chaotic ($\alpha > 1$). This is actually what we observe, at least for the hyperbolic tangent nonlinearity, and less so for the sigmoid.

The context-dependent decision task on the other hand, requires a different type of computation: the reservoir should be capable of temporal pattern recognition. Previous studies have shown that the separability of the reservoir -or its ability to recognize or discriminate among multiple external stimuli- is rather supported by the presence of chaos. This can explain the observed increase in performance as global network dynamics transition from stable to chaotic in the context-dependent decision making task, regardless of the type of local dynamics. Altogether, these results demonstrate how different types of computations are supported by different types of global network dynamics.

We have modified the *Tutorial* section of the manuscript to include this explanation. The new version reads:

“Results in Fig. 4 suggest that both local and global network dynamics have an impact on task performance. At the local level, both tasks benefit from having a hyperbolic tangent activation function, compared to the sigmoid. However, dependence of task performance on global network dynamics varies from one task to the other. In the perceptual decision making task, a choice must be made based on the time integration of two past stimuli, which means that a temporal memory is required. Because stability enforces memory in the reservoir, computations required in the perceptual decision making task should take advantage of stable network dynamics [32, 42, 43, 50]. This is indeed the case: if the local nonlinearity is hyperbolic tangent, a decrease in performance from stable ($\alpha < 1$) to chaotic ($\alpha > 1$) dynamics is observed (pink lines in bottom panel on the left column of Fig. 4). If the local nonlinearity is a sigmoid, however, the reservoir does not show a strong dependence with respect to global network dynamics (green lines in bottom panel on the left column of Fig.4). In contrast, in

the context-dependent decision making task, a binary perceptual discrimination must be made, and hence the reservoir must learn to differentiate between two temporal patterns. Because chaotic dynamics contribute to the separability property of a reservoir [32, 42, 43, 50], performance in this task should be enhanced by the presence of chaos. This is observed by an increase in performance as global network dynamics transition from stable to chaotic (pink and green lines in bottom panel on the right column of Fig.4). Even though this is observed for both types of local nonlinearities - i.e., hyperbolic tangent and sigmoid - the effects are stronger for the hyperbolic tangent type. As expected, the effect of local and global network dynamics on task performance depends on the type of computations required by the task at hand.”

[32] Maass, W., Natschläger, T., & Markram, H. (2002). Real-time computing without stable states: A new framework for neural computation based on perturbations. *Neural computation*, 14(11), 2531-2560.

[42] Legenstein, R., & Maass, W. (2007). Edge of chaos and prediction of computational performance for neural circuit models. *Neural networks*, 20(3), 323-334.

[43] Legenstein, R., & Maass, W. (2007). What makes a dynamical system computationally powerful. *New directions in statistical signal processing: From systems to brain*, 127-154.

[50] Bertschinger, N., & Natschläger, T. (2004). Real-time computation at the edge of chaos in recurrent neural networks. *Neural computation*, 16(7), 1413-1436.

References used in the response:

[1] Legenstein, R., & Maass, W. (2007). What makes a dynamical system computationally powerful. *New directions in statistical signal processing: From systems to brain*, 127-154.

5. Legends and captions of figures are too small.

We agree with the Reviewer that some of the legends of the figures are too small. We have therefore increased their font size, improving the readability of the figures.

Reviewer #3 (Remarks to the Author):

The manuscript presents the `conn2res` toolbox for connectome constrained reservoir computing. It is very well written, including a very accessible presentation of the ideas behind reservoir computing and its history in computer science and neuroscience. I have explored the code which is implemented in a very clean but accessible way that should maximise its utility in the computational neuroscience community. I have some suggestions about additional topics/issues that should be addressed and about some simple additional tutorial examples that would extend the reach/accessibility of the toolbox.

We thank the Reviewer for the comments and suggestions. Below we provide a detailed answer to each of the Reviewer's concerns.

The emphasis of the work is on connectome based models. Given the increasing recognition of geometric constraints on propagation of activity along the cortex, how would local geometric influences (e.g., dynamics driven by a mixture of geodesically proximal and white-matter connectome activity) be incorporated into the models. Some discussion of this/computational tradeoffs would be useful.

There are several ways in which geometric constraints can be incorporated into our or other similar neuroAI frameworks. For example, geometry can be considered by adding time-delays or conduction velocities to the transmission of information through white matter connections, as is commonly done in neural mass models (e.g. Deco et al., 2011, Nature Reviews Neuroscience). An alternative approach is the recently proposed concept of spatially-embedded recurrent neural networks (esRNNs; Achterberg et al., 2023, Nature Machine Intelligence). These are trainable RNNs, confined within a 3D Euclidean space, whose weight learning is subject to biological optimization processes such as the minimization of wiring costs or the maximization of inter-regional communicability. We have added a new paragraph on this point to the *Concluding remarks* section:

“The RC paradigm can also be adapted to jointly study the influence of network wiring and spatial embedding on computation. Namely, the placement of connections in the brain is subject to numerous material, energetic and spatial constraints, a factor that is often overlooked in classical paradigms that focus exclusively on network topology [94]. Right now the models included in the `conn2res` toolbox do not explicitly take into account spatial embedding but they can be readily adapted to do so. One way is to introduce conduction delays that are proportional to inter-regional connection lengths or geodesic distances over the cortical surface [95, 96]. Another interesting and slightly different approach that incorporates geometric constraints is the recently introduced concept of spatially-embedded recurrent neural networks (seRNNs) [97]. These are recurrent networks with adaptive weights, confined within a 3D Euclidean space, whose learning is constrained by biological optimization processes, like the

minimization of wiring costs or the optimization of inter-regional communicability, in addition to the maximization of computational performance. When the pruning of the network is guided by these biological optimization principles, the resulting network architecture displays characteristic features of biological brain networks, such as modular structure with a small-world topology, and the emergence of functionally specialized regions that are spatially co-localized and implement an energetically-efficient, mixed-selective code [97, 98].”

[94] Pang, J. C., Aquino, K. M., Oldehinkel, M., Robinson, P. A., Fulcher, B. D., Breakspear, M., & Fornito, A. (2023). Geometric constraints on human brain function. *Nature*, 1-9.

[95] Deco, G., Jirsa, V., McIntosh, A. R., Sporns, O., & Kötter, R. (2009). Key role of coupling, delay, and noise in resting brain fluctuations. *Proceedings of the National Academy of Sciences*, 106(25), 10302-10307.

[96] Deco, G., Jirsa, V. K., & McIntosh, A. R. (2011). Emerging concepts for the dynamical organization of resting-state activity in the brain. *Nature reviews neuroscience*, 12(1), 43-56.

[97] Achterberg, J., Akarca, D., Strouse, D. J., Duncan, J., & Astle, D. E. (2023). Spatially embedded recurrent neural networks reveal widespread links between structural and functional neuroscience findings. *Nature Machine Intelligence*, 1-13.

[98] Rigotti, M., Barak, O., Warden, M. R., Wang, X. J., Daw, N. D., Miller, E. K., & Fusi, S. (2013). The importance of mixed selectivity in complex cognitive tasks. *Nature*, 497(7451), 585-590.

There are other connectome-based reservoir-like models which haven't been mentioned in the manuscript (e.g., Hellyer et al, 2017 Plos Comp Biol). They also highlight other issues which could be touched on in the manuscript: e.g., brain-environmental feedback loops and adaptive, homeostatic mechanisms (see also Hellyer et al, 2016 Neuroimage). These may be relevant especially in the context of efficient/localised method for maintaining reservoirs near criticality; also potentially very relevant for modelling changes across time/disease, ontogeny/phylogeny. See also, Falandays et al 2023 Cognitive Neurodynamics for recent work (behavioural) on self-organisation to perform a range of tasks based on reservoir local homeostasis.

Thanks for pointing out Hellyer and colleagues' and Falandays and colleagues' works. They indeed deal with other aspects such as environmental feedback loops and homeostatic mechanisms, which could be operationalized using reservoir computing. We have therefore included and briefly discussed these works in the *Concluding remarks* section of the manuscript:

“[...]. More broadly the cortex, which is typically studied in these models, is part of a wider network of the central nervous system that is embedded in a perpetually changing environment. The RC paradigm can accommodate this “embodied”

view of the brain. Specifically, RC models can include adaptive homeostatic mechanisms that regulate brain-environmental feedback loops to ensure that reservoirs are maintained in a desired dynamical state such as criticality [99-101].”

[99] Hellyer, P. J., Clopath, C., Kehagia, A. A., Turkheimer, F. E., & Leech, R. (2017). From homeostasis to behavior: Balanced activity in an exploration of embodied dynamic environmental-neural interaction. *PLoS computational biology*, 13(8), e1005721.

[100] Hellyer, P. J., Jachs, B., Clopath, C., & Leech, R. (2016). Local inhibitory plasticity tunes macroscopic brain dynamics and allows the emergence of functional brain networks. *NeuroImage*, 124, 85-95.

[101] Falandays, J. B., Yoshimi, J., Warren, W. H., & Spivey, M. J. (2023). A potential mechanism for Gibsonian resonance: Behavioral entrainment emerges from local homeostasis in an unsupervised reservoir network. *Cognitive Neurodynamics*, 1-24.

The tutorial workbook is very clean and accessible. Many users will just run the code with existing functions. However, it might be nice to have an example workbook of writing and running bespoke models (e.g., other reservoirs/connectome approaches etc). This would make it much easier for users to adapt the code for a wider range of purposes.

In line with the Reviewer’s suggestion, we have extended the *Tutorial* section to include a total of four study-cases. The new *Tutorial* section is now divided in two parts: *Example 1: Toolbox components* and *Example 2: Applications*. The first part consists of a detailed step-by-step, toy example that illustrates the generic *conn2res* workflow. The second part consists of three new applied examples that show how the *conn2res* toolbox can be used to address different types of inferences using various reservoirs/connectome approaches, as well as other behavioral paradigms. The new examples are accompanied by ready-to-use code that can be easily customized by the user. The scripts have been added to the example workbook of the toolbox (<https://github.com/netneurolab/conn2res/tree/master/examples>). The new *Example 2: Applications* subsection includes a new figure (Figure 5). The new *Example 2: Applications* subsection in the *Tutorial* section reads:

“Example 2: Applications

In the second part of the tutorial we show how the toolbox can be applied to address three specific biological questions. First, we quantify the memory capacity of human white matter connectomes reconstructed using diffusion weighted MRI (n=66; [85]). Here we use subcortical regions as input nodes and cortical regions as readout nodes. We then ask whether memory capacity specifically depends on the topological organization of the brain, rather than low-level features such as density or degree-sequence. To address this question, we compare memory capacity in a group-level, empirical connectome against memory capacity in a population of 500 randomly rewired null connectomes with

preserved density and degree sequence [55, 88]. Fig. 5a shows that at criticality ($\alpha = 1$), the memory capacity of empirical brain networks is significantly greater than in rewired nulls, suggesting that the topology of the human brain confers computational function [35].

In the previous example we focused on global computational capacity and showed that it relates to global network topology. For the second example, we demonstrate how the toolbox can be used to make inferences about regional heterogeneity or specificity for computational capacity. To address this question we implement the perceptual decision making task on a single subject-level, connectome-informed reservoir. We stratify cortical nodes according to their affiliation with the canonical intrinsic networks [86]. Specifically, we use brain regions in the visual network as input nodes, and the remaining networks separately as a readout module each to quantify task performance. Fig. 5b shows prominent differentiation in performance depending on which network is chosen as the readout module. Interestingly, the two modules with the greatest performance are the default mode and somatomotor networks, consistent with the notion that perceptual decision making involves integration of sensory inputs, comparison with internal goals, and formulation of an appropriate motor response [93]. Collectively, these results demonstrate how connectome-based reservoir computing can be used to make inferences about the computational capacity of anatomically circumscribed neural circuits.

For the final example, we show how the toolbox can be applied to comparative questions in which a researcher seeks to compare networks. In this example, we implement connectomes reconstructed from four different species: fruit fly [89], mouse [90], rat [91] and macaque [92]. As in the first example, we compare memory capacity in each empirical connectome with a population of 500 rewired null networks. Fig. 5c shows that, despite differences in brain size, connectome resolution and reconstruction technique, the four model organism connectomes show a similar dependence on dynamics. Importantly, as with the human connectome, peak memory capacity is significantly greater in the empirical connectomes compared to the rewired nulls, except for the fruit fly, suggesting that this principle is potentially ubiquitous across nervous systems.”

a | global network organization

b | regional specificity

c | cross-species comparison

Figure5. **Applied examples** | (a) Inferences on global network organization: a reservoir informed by a human consensus connectome ($n=66$ subjects; diffusion-weighted MRI [85]) was implemented to perform a memory capacity task. Subcortical regions were used as input nodes while cortical regions were used as readout nodes. (Left) In all cases ---panels (a) to (c)---, global dynamics were tuned to transition from stable ($\alpha < 1$) to chaotic ($\alpha > 1$). (Right) The performance of the empirical network was compared against the performance of a family of 500 rewired nulls that preserve network density and degree sequence [55, 88]. At criticality ($\alpha_{crit} = 1$), empirical networks perform significantly better than rewired nulls ($p = 0.002$). (b) Anatomical inferences: a single, subject-level connectome-informed reservoir was implemented to perform a perceptual decision making task (500 iterations). (Right) Cortical regions were stratified according to the intrinsic network they belong to. Brain regions in the visual network were used as input nodes; the remaining intrinsic networks were used as separate readout modules each to quantify task performance. (Left) Across a wide range of α values ($\alpha > 0.5$), all intrinsic networks display significantly different behavior (one-way ANOVA $F = 1143.50$, $p < 0.002$ at $\alpha_{crit} = 1$), thus suggesting functional specialization across these networks. (c) Cross-species

comparison: we implemented four distinct reservoirs, each informed by the connectome of a different model organism: fruit fly [89], mouse [90], rat [91] and macaque [92]. (Left) Connectome-informed reservoirs were trained to perform a memory capacity task. Sensory areas were used as input nodes while motor areas were used as readout nodes. As in (a), empirical networks were compared to a family of 500 rewired nulls [88]. At peak memory capacity, biologically-informed connectomes perform significantly better than rewired nulls, except for the fruit fly (fruit fly: $p = 0.11$ at $\alpha_{\text{peak}} = 0.9$, mouse: $p < 0.002$ at $\alpha_{\text{peak}} = 1.05$, rat: $p < 0.002$ at $\alpha_{\text{peak}} = 1.15$, macaque: $p < 0.002$ at $\alpha_{\text{peak}} = 1.15$).

[35] Suárez, L. E., Richards, B. A., Lajoie, G., & Misic, B. (2021). Learning function from structure in neuromorphic networks. *Nature Machine Intelligence*, 3(9), 771-786.

[55] Váša, F., & Mišić, B. (2022). Null models in network neuroscience. *Nature Reviews Neuroscience*, 23(8), 493-504.

[85] Griffa, A., Alemán-Gómez, Y., & Hagmann, P. (2019). Structural and functional connectome from 70 young healthy adults [data set]. *Zenodo*.

[86] Yeo, B. T., Krienen, F. M., Sepulcre, J., Sabuncu, M. R., Lashkari, D., Hollinshead, M., ... & Buckner, R. L. (2011). The organization of the human cerebral cortex estimated by intrinsic functional connectivity. *Journal of neurophysiology*.

[88] Maslov, S., & Sneppen, K. (2002). Specificity and stability in topology of protein networks. *Science*, 296(5569), 910-913.

[89] Chiang, A. S., Lin, C. Y., Chuang, C. C., Chang, H. M., Hsieh, C. H., Yeh, C. W., ... & Hwang, J. K. (2011). Three-dimensional reconstruction of brain-wide wiring networks in *Drosophila* at single-cell resolution. *Current biology*, 21(1), 1-11.

[90] Rubinov, M., Ypma, R. J., Watson, C., & Bullmore, E. T. (2015). Wiring cost and topological participation of the mouse brain connectome. *Proceedings of the National Academy of Sciences*, 112(32), 10032-10037.

[91] Bota, M., Sporns, O., & Swanson, L. W. (2015). Architecture of the cerebral cortical association connectome underlying cognition. *Proceedings of the National Academy of Sciences*, 112(16), E2093-E2101.

[92] Modha, D. S., & Singh, R. (2010). Network architecture of the long-distance pathways in the macaque brain. *Proceedings of the National Academy of Sciences*, 107(30), 13485-13490.

[93] Heekeren, H. R., Marrett, S., & Ungerleider, L. G. (2008). The neural systems that mediate human perceptual decision making. *Nature reviews neuroscience*, 9(6), 467-479.

Minor points:

"Given the sensitivity of Hebbian-like learning mechanisms to temporal relations [90–93], the synchronization of oscillatory activity can favour the formation of synaptic connections and hence can be used for establishing associations. In this way, as posited by the “binding by synchrony” hypothesis [94–96], synchronous oscillations act

as a binding mechanism for distributed nodes in the network that represent features of the same cognitive object, thus serving as a general code of related-ness [96]. Likewise, phase shifts can lead to rapid re-versals of the direction of information flow, thus serving as gating mechanisms for information transfer according to the “communication through coherence” hypothesis” I didn’t understand the relevance of this paragraph.

Even though the intention of the manuscript is not necessarily to propose reservoir computing as a model for how the brain computes, it is still interesting to consider how different dynamical phenomena (e.g., the synchronization of oscillatory activity) that take place in recurrent neural networks, or *reservoirs*, could actually serve as mechanisms that support learning and computation in biological brains. This paragraph was in *The brain as a reservoir* review section of the original manuscript. Given its length and content -also following Reviewers’ 1 and 2 advice, we have removed this section in the revised version of the manuscript and hence this paragraph does not appear anymore.

"In the future, continuous-time nonlinear dynamical models will be implemented" Some expansion of this would be useful.

We agree with the Reviewer’s suggestion and we have taken several actions to expand on this point. First, we have implemented a spiking neuron model as an example of a continuous-time, nonlinear dynamical model, i.e., a model governed by differential equations (as opposed to models governed by difference equations as is the case of the discrete-time dynamical models already included in the *conn2res* toolbox), and second, we have revised the manuscript to clarify this point. Specifically, the second paragraph of *The conn2res toolbox* section has been modified to reflect these changes:

“[...] **(v) reservoir local dynamics**: the type of dynamics governing the activation of the reservoir’s units. Local dynamics can be split into two categories: discrete-time, governed by difference equations and continuous-time, based on differential equations. The former category includes both linear and non-linear artificial neuron models with activation functions such as ReLU, leaky ReLU, sigmoid, and hyperbolic tangent, whereas the latter category includes a non-linear spiking neuron model. The underlying leaky-integrate-and-fire neuron model is based on the framework proposed in [80], with model parameters from [81] and implements a double-exponential synaptic filter for the presynaptic spike trains. This flexible model provides higher biological plausibility, notably allowing for user-specified heterogeneous synaptic time constants, artificial stimulations/inhibitions, and interneuron-specific connectivity constraints [80, 82]; [...]”

[80] Kim, R., Li, Y., & Sejnowski, T. J. (2019). Simple framework for constructing functional spiking recurrent neural networks. *Proceedings of the national academy of sciences*, 116(45), 22811-22820.

[81] Nicola, W., & Clopath, C. (2017). Supervised learning in spiking neural networks with FORCE training. *Nature communications*, 8(1), 2208.

[82] Song, H. F., Yang, G. R., & Wang, X. J. (2016). Training excitatory-inhibitory recurrent neural networks for cognitive tasks: a simple and flexible framework. *PLoS computational biology*, 12(2), e1004792.

For details of the newly implemented spiking neuron model please refer to our response to Reviewer 1's third comment on pages 11 to 13.

REVIEWERS' COMMENTS

Reviewer #1 (Remarks to the Author):

Now, the manuscript after substantial revision has addressed my major concerns. Before the final acceptance, I have the following minor suggestion.

The authors mentioned the works of RCs in different physical types, including the analog circuits [102-105]. I would like to recommend the following advances in developing efficient and useful frameworks for RCs particularly using time delays or similar settings.

1. Phys. Rev. Res. 5, L022041 (2023) <https://doi.org/10.1103/PhysRevResearch.5.L022041>
2. Sci. Rep. 10, 21794 (2020). <https://doi.org/10.1038/s41598-020-78725-0>
3. Phys. Rev. X 7, 011015 (2017). <https://doi.org/10.1103/PhysRevX.7.011015>

Reviewer #2 (Remarks to the Author):

The authors made significant revisions to the original manuscript, and I believe that the current version addresses all the concerns highlighted in my previous comments. Specifically, restructuring the manuscript by removing an entire section and adding application examples to the Tutorial section not only eliminated the review/tutorial-like format but also highlighted its major contributions. I think this version of the paper was strengthened by the authors' hard work, and I recommend the current manuscript for publication in Nature Communications.

Reviewer #3 (Remarks to the Author):

The reviewers have satisfactorily addressed my comments.

Dear Reviewers,

Thanks again for the constructive feedback on our second submission. Following your suggestion, we have revised the manuscript. Here, we respond to the Reviewers' comments in detail. The Reviewers' comments appear in **bold font** and our response in regular font.

REVIEWERS' COMMENTS

Reviewer #1 (Remarks to the Author):

Now, the manuscript after substantial revision has addressed my major concerns. Before the final acceptance, I have the following minor suggestion.

The authors mentioned the works of RCs in different physical types, including the analog circuits [102-105]. I would like to recommend the following advances in developing efficient and useful frameworks for RCs particularly using time delays or similar settings.

1. Phys. Rev. Res. 5, L022041 (2023) <https://doi.org/10.1103/PhysRevResearch.5.L022041>
2. Sci. Rep. 10, 21794 (2020). <https://doi.org/10.1038/s41598-020-78725-0>
3. Phys. Rev. X 7, 011015 (2017). <https://doi.org/10.1103/PhysRevX.7.011015>

We thank the Reviewer for the positive feedback. Following the Reviewer's minor comment, we have introduced the suggested works in the *Concluding remarks* section:

"[...] The development of physical reservoir systems has been accompanied by advances in more efficient and effective RC frameworks, for instance by including time delays [141-143]. [...]"

[141] Duan, X. Y., Ying, X., Leng, S. Y., Kurths, J., Lin, W., & Ma, H. F. (2023). Embedding theory of reservoir computing and reducing reservoir network using time delays. *Physical Review Research*, 5(2), L022041.

[142] Sakemi, Y., Morino, K., Leleu, T., & Aihara, K. (2020). Model-size reduction for reservoir computing by concatenating internal states through time. *Scientific reports*, 10(1), 21794.

[143] Larger, L., Baylón-Fuentes, A., Martinenghi, R., Udaltsov, V. S., Chembo, Y. K., & Jacquot, M. (2017). High-speed photonic reservoir computing using a time-delay-based architecture: Million words per second classification. *Physical Review X*, 7(1), 011015.

Reviewer #2 (Remarks to the Author):

The authors made significant revisions to the original manuscript, and I believe that the current version addresses all the concerns highlighted in my previous comments.

Specifically, restructuring the manuscript by removing an entire section and adding application examples to the Tutorial section not only eliminated the review/tutorial-like format but also highlighted its major contributions. I think this version of the paper was strengthened by the authors' hard work, and I recommend the current manuscript for publication in Nature Communications.

We thank the Reviewer for the positive feedback.

Reviewer #3 (Remarks to the Author):

The reviewers have satisfactorily addressed my comments.

We thank the Reviewer for the positive feedback.